# Bio-Inspired Self-Supervised Learning for Wrist-worn Accelerometer Data

Prithviraj Tarale [1]   Kiet Chu [1]   Abhishek Varghese [1]   Kai-Chun Liu [2]   Maxwell A. Xu [3]   Mohit Iyyer [4]
Sunghoon I. Lee [1]

## Abstract

Wearable accelerometers enable large-scale health monitoring, yet learning robust human-activity representations has been constrained by scarce labeled data. While self-supervised learning offers a remedy, existing methods treat sensor streams as unstructured time series, overlooking the underlying biological structure of human movement, a factor we argue is critical for effective Human Activity Recognition (HAR). We introduce a novel tokenization strategy grounded in the *submovement theory* of motor control, which posits that continuous wrist motion is composed of elementary basis functions called submovements. We define our token as the *movement segment*, a computationally tractable unit of motion composed of a finite sequence of submovements. By pretraining a Transformer encoder via masked reconstruction of these tokens, we shift the learning focus from local waveform morphology to high-level structural and temporal organization. Pretrained on the NHANES corpus ($\approx$ 28k hours; $\approx$ 11k participants), our representations outperform strong wearable SSL baselines across six subject-disjoint HAR benchmarks. Code and pretrained weights are available at https://prithvitarale.github.io/biopm-site/.

## 1. Introduction

The ubiquity of wrist-worn inertial sensors enables transformative healthcare applications, from fitness and sleep tracking (Fu et al., 2020; Levy et al., 2023) to personalized medicine (Dunn et al., 2018; Small et al., 2019). However,

[1]College of Information and Computer Sciences, University of Massachusetts, Amherst, United States [2]Stevens Institute of Technology, Hoboken, United States [3]Google Health, Seattle, United States [4]Department of Computer Science, University of Maryland, College Park, United States. Correspondence to: Prithviraj Tarale <ptarale@umass.edu>.

*Proceedings of the 43$^{rd}$ International Conference on Machine Learning*, Seoul, South Korea. PMLR 306, 2026. Copyright 2026 by the author(s).

realizing their full potential requires robust representations that can recognize a wide spectrum of human activities from raw data—spanning primitive upper-limb actions like reaching-to-grasp (Wang et al., 2025), simple behaviors such as walking or sleeping (Kontaxis et al., 2024), and complex functional activities like cooking or cleaning (Shoaib et al., 2016). Unfortunately, progress in Human Activity Recognition (HAR) has been hampered by the scarcity of large-scale labeled datasets, primarily due to the prohibitive costs of manual annotation (Haresamudram et al., 2025).

To mitigate this label scarcity, recent research has pivoted toward self-supervised learning (SSL) to build representations from vast volumes of unlabeled wrist-worn accelerometer data, designed to transfer to limited labeled datasets (Yuan et al., 2024; Xu et al., 2025a;b). Typically, these models focus on learning features that characterize local time-series morphology within fixed-length sliding windows via unsupervised objectives, such as contrastive learning (Chen et al., 2020; Zhang et al., 2022), masked reconstruction (Narayanswamy et al., 2024), or augmentation prediction (Yuan et al., 2024). However, a fundamental limitation persists: they treat accelerometer data as unstructured time series and optimize objectives over fixed-length windows with arbitrary boundaries, rather than meaningful movement units. This is a significant drawback. Prior research demonstrates that tokenization reflecting a domain's underlying structure (e.g., words in natural language) provides a critical inductive bias for modeling compositional structure, substantially improving downstream performance (Rajaraman et al., 2024; Gastaldi et al., 2024; Schmidt et al., 2024). Without meaningful tokens, SSL may exhaust its capacity attempting to recover the underlying structure from arbitrary fragments of data, rather than learning high-level relationships between coherent events. Therefore, transitioning from arbitrary time windows to structurally meaningful tokens represents a critical step toward unlocking the full potential of wearable SSL.

In this work, we bridge this gap by introducing a bio-inspired tokenization of wrist-accelerometer data, enabling sequence modeling over meaningful movement units and their temporal relations via self-supervised learning. Our approach is grounded in the *submovement theory of motor control*, which posits that continuous, complex wrist

movements are constructed from the superposition of elementary units known as submovements (Rohrer et al., 2004). These submovements act as the fundamental building blocks of observed wrist motion, analogous to how characters or phonemes serve as the building blocks of written or spoken language. Leveraging this theoretical framework, we propose a novel token definition: the *movement segment*. Defined as a sub-second unit composed of a finite number of submovements, we posit that this segment functions as the equivalent of a "word" in natural language. The primary hypothesis of this work is that these movement segments capture useful aspects of the organizational structure of human activity, allowing the model to look beyond simple waveform morphology.

Leveraging this framework, we also introduce **Bio-PM** (Bio-inspired Tokenization-based Pretrained Model), an open pretrained encoder designed to model the temporal dependencies between movement segments. Bio-PM is pretrained on NHANES, a large-scale public wrist accelerometer corpus comprising approximately 28,000 hours of data from 11,000 participants (Centers for Disease Control and Prevention (CDC), 2010). Empirically, Bio-PM achieves the strongest performance among the controlled SSL baselines we evaluate, improving macro-F1 scores by an average of 6% (range: 3–12%) across six public HAR benchmarks.

Our primary contributions are:

1. **Bio-inspired Tokenization for wrist accelerometer.** We introduce a scalable tokenization strategy that segments continuous accelerometer signals into meaningful movement units for sequence modeling.
2. **Contextual Representation Learning.** We present Bio-PM, a Transformer-based encoder pretrained via masked movement-segment reconstruction to capture the compositional structure of human activity.
3. **Data-efficient transfer from large-scale pretraining.** We show that movement segment-based pretraining improves label efficiency over SSL baselines.

## 2. Related Work

Wearable HAR has increasingly leveraged SSL to reduce reliance on labeled data (Haresamudram et al., 2022). Most efforts focus on wrist-worn accelerometers because the wrist is the most socially acceptable site for long-term monitoring (Yuan et al., 2024), and omitting gyroscopes significantly extends battery life. These practical choices have enabled the large-scale unlabeled corpora (e.g., NHANES, UK Biobank) required to make SSL viable.

Previously proposed SSL methodologies generally fall into three dominant families of objectives: (i) augmentation prediction, which trains models to recognize the presence or type of transformations applied to the signal (Saeed et al., 2019; Yuan et al., 2024); (ii) masked reconstruction, which trains models to impute missing data points within an accelerometer stream (Haresamudram et al., 2020; Narayanswamy et al., 2024); and (iii) contrastive learning, which teaches the model to recognize that different distorted versions of the same signal share the same underlying identify (Chen et al., 2020; Xu et al., 2025a), including variants like Time–Frequency Consistency (TF-C) that match features across a signal's temporal and spectral domains (Zhang et al., 2022).

Each family presents distinct challenges. Augmentation prediction relies on the assumption that signal-level transformations (e.g., temporal reversal or chunk-and-permute) reflect realistic sensor variability. Unfortunately, unintended data artifacts caused by these potentially unrealistic transformations—such as the unnatural, jagged transitions created by chopping and shuffling a signal—often provide models with an easy shortcut. As a result, the model achieves high pretraining accuracy by exploiting these artificial errors rather than learning the actual underlying patterns of human movement. Masked reconstruction provides dense supervision, but reconstructing raw sensor values can often be trivially solved via local interpolation, failing to capture broader contextual dependencies. Finally, while contrastive objectives—including the highly competitive TF-C baseline—often transfer well, their success depends entirely on exactly how the training data is augmented and grouped. Because these models are forced to find similarities across heavily altered data, they tend to learn broad, 'big-picture' patterns rather than the fine-grained, subtle details of human movement.

Beyond these method-specific challenges, all three SSL families share a more fundamental limitation: they process the signal as an unstructured series of raw data points. In contrast, the proposed bio-inspired tokenization strategy diverges fundamentally by parsing the enclosed signal into a sequence of structurally meaningful motion events. To demonstrate that this lack of internal structure is a critical bottleneck, we isolate and evaluate tokenization as a distinct, foundational design axis in wearable SSL.

## 3. Methods

### 3.1. Biological Prior: The Submovement Theory

The proposed tokenization draws on the *submovement theory* of motor control (Krebs et al., 1999; Elliott et al., 2001; Rohrer et al., 2004; Hogan & Sternad, 2012; Miranda et al., 2018). The theory treats the hand/wrist as the upper-limb's end-effector and posits that its continuous motion can be described as a superposition of discrete, bell-shaped kinematic units known as *submovements*. For each local coordinate

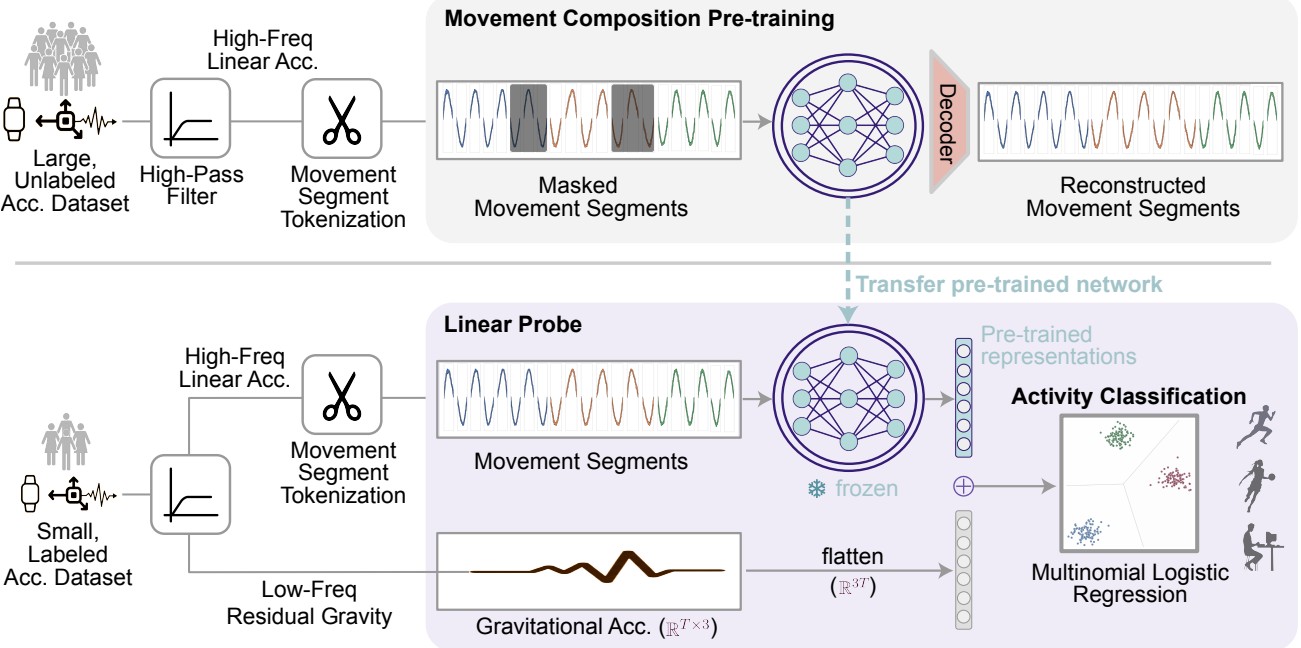

*Figure 1.* Bio-PM Representation Learning. We (i) tokenize accelerometry into movement-aligned segments, (ii) pretrain by modeling temporal relation across segments with a Transformer under masked reconstruction, and (iii) transfer the frozen encoder to downstream HAR for linear probing.

axis of the wrist $i \in \{x, y, z\}$, velocity is expressed as

$$v_i(t) = \sum_{k=1}^{K} \bar{v}_{ik} \cdot \sigma(t \mid \tau_k, d_k), \qquad (1)$$

where $K$ is the number of submovements. Each submovement $k$ can be defined by a normalized bell-shaped basis function $\sigma(\cdot)$ with an onset $\tau_k$ and duration $d_k$ that are shared across the three axes, while the peak velocity $\bar{v}_{ik}$ remains independent for each axis.

The canonical bell-shaped profile originates from observations of point-to-point reaching tasks—the most fundamental form of goal-directed upper-limb movement—where the hand moves between two positions, starting and ending at rest (Guigon et al., 2007). Crucially, this profile exhibits *shape invariance*: although parameters such as duration $d_k$ and peak velocity $\bar{v}_{ik}$ scale with movement distance and speed, the underlying normalized shape $\sigma(\cdot)$ remains constant (Flash & Hogan, 1985). Submovement theory generalizes this principle to complex behaviors, modeling continuous velocity profiles as sequences of partially overlapping sublinear movements (Hogan & Sternad, 2012).

We seek a tokenization that captures the compositional structure of human movement described by submovement theory while remaining computationally tractable for continuous wearable accelerometer streams. Although the *submovement* is the atomic unit of upper-limb movements, using it directly as a token introduces a critical limitation. Estimat-

ing the submovement parameters in Equation (1) requires iterative fitting procedures (Rohrer & Hogan, 2003; 2006), which are prohibitively expensive to run continuously at scale.

We therefore adopt the *movement segment* (Daneault et al., 2023) as our guiding unit of tokenization. Defined as the interval between successive velocity zero-crossings (Figure 2 top), a movement segment encapsulates one or more submovements bounded by moments of rest. Conceptually, we view movement segments as analogous to "words," whereas individual submovements function as "phonemes" or "letters"—essential for composition, but rarely meaningful in isolation. Crucially, we apply this segmentation independently to each local coordinate axis. This bypasses the need to perfectly align submovement onsets and durations across all three spatial dimensions, significantly reducing computational complexity while enhancing robustness to noisy accelerometer data. We posit that the spatiotemporal organization of these segments across the three axes captures the underlying composition of submovements, yielding higher-level contextual patterns that sequential models can directly exploit. However, tokenizing directly in the velocity domain poses significant practical hurdles; because accelerometers measure acceleration, velocity must be obtained by numerical integration (e.g., Simpson's rule). This process not only increases computational overhead but also introduces artifacts like integration drift, which degrade signal fidelity over continuous sensor streams.

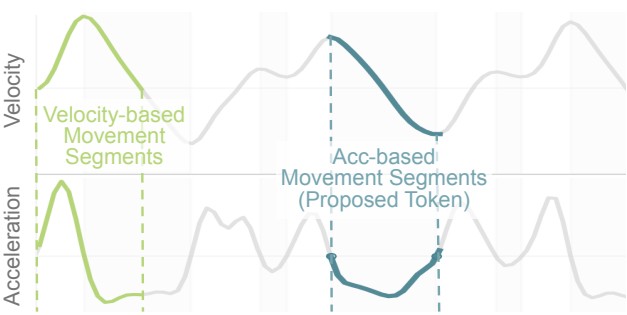

*Figure 2.* Illustration of the proposed tokenization strategy, which uses "type 2" movement segments defined via acceleration zero-crossings.

## 3.2. Proposed Tokenization of Wearable Accelerometer Data

To circumvent these complications, we define token boundaries at acceleration zero-crossings (Figure 2 bottom), formally referred to as "type 2" movement segments (Simo et al., 2014; Nunes et al., 2025; Noy et al., 2025). This strategy is kinematically justified: a bell-shaped velocity profile manifests as a biphasic acceleration profile, where the peak velocity aligns with a zero-crossing in acceleration. Furthermore, this choice is supported by clinical research demonstrating that the morphology and temporal organization of these segments encode critical markers of motor impairment in neurological conditions, such as stroke (Simo et al., 2014), Parkinson's disease (Noy et al., 2025), and Huntington's disease (Nunes et al., 2025). The proven sensitivity of these segments to underlying motor phenotypes corroborates our hypothesis that they capture the latent compositional structure of human movement, planned and controlled by the central nervous system.

## 3.3. Signal Preprocessing

The biological premise and tokenization strategy described above apply specifically to linear acceleration of the upper limb. Raw accelerometer data, however, inherently combines this linear component with gravitational acceleration. To estimate linear kinematics, we apply a 0.5 Hz Butterworth high-pass filter to each axis, a standard practice in inertial sensing (Van Hees et al., 2013; Nunes et al., 2025). The resulting high-frequency component is then tokenized into movement segments by detecting zero-crossings in each filtered axis (Appendix Figure 6).

Importantly, we retain the residual low-frequency component, as it carries important semantic value. Because digital filtering inherently leaves a residual mix of static gravity and slow rotational movements, this component contains essential postural context (e.g., distinguishing sitting from lying down). We detail how both the high- and low-frequency components are integrated during model training and downstream evaluation in Sections 3.4 and 3.5, respectively.

## 3.4. Self-Supervised Pretraining Pipeline

To learn representations of wrist accelerometers that capture the temporal organization of movement segments, we pretrained a Transformer encoder using a masked reconstruction objective. In this section, we describe (i) accelerometer windowing and movement-segment tokenization, (ii) CNN segment encoding, (iii) Transformer sequence modeling, and (iv) masked segment reconstruction.

**Windowing and tokenization.** We partition the gravity-reduced accelerometer stream into non-overlapping 10 s windows, $\mathcal{X}_j \in \mathbb{R}^{T \times 3}$, where $j$ indexes windows, and $T$ is the number of samples per window. For each axis $A \in \{x, y, z\}$, we detect consecutive sign changes (zero-crossings) in $\mathcal{X}_j^{(:,A)}$ and define a movement segment as the samples between successive zero-crossings. Let $\mathcal{S}_j^{i,A}$ denote the $i^{\text{th}}$ segment on axis $A$ within window $j$. To suppress spurious zero-crossings induced by sensor noise, we apply a two-stage hysteresis filter. This filter merges any insignificant segment—defined as having a duration shorter than 50 ms or a peak amplitude below 0.01 g—into the subsequent segment that meets these thresholds. By filtering out these micro-fluctuations, we allow the model to focus on the systematic organization of prominent, semantically meaningful movement patterns (detailed in Appendix F.2).

We resample each segment $\mathcal{S}_j^{i,A}$ to a fixed length $L=32$ using linear interpolation, yielding $\bar{\mathcal{S}}_j^{i,A} \in \mathbb{R}^L$. We also retain segment metadata: (i) duration of each sample $D_j^{i,A}$ to preserve original temporal fidelity, (ii) midpoint time $t_j^{i,A}$ within the zero-crossing-window for position encoding, and (iii) axis identifier $A$ to preserve spatial fidelity.

**Movement-segment encoder (CNN).** Because the local waveform within a movement segment encodes kinematic structure (e.g. peaks, asymmetry, and oscillation rate), we apply a 1D CNN to each resampled segment $\bar{\mathcal{S}}_j^{i,A} \in \mathbb{R}^L$ to produce an embedding $h_j^{i,A} \in \mathbb{R}^{60}$. We form the token representation by concatenating (i) the CNN embedding, (ii) a learned 3D axis embedding $e(A)$, and (iii) its duration $D_j^{i,A}$ (scalar):

$$z_j^{i,A} = \left[ h_j^{i,A}, e(A), D_j^{i,A} \right]. \tag{2}$$

This construction yields tokens $z_j^n \in \mathbb{R}^{64}$ (60 CNN + 3 axis + 1 duration), chosen based on upstream model size analysis (Appendix Table 4). We then merge tokens from all three axes and sort them by midpoint time to obtain a single token sequence $z_j^n{}_{n=1}^{N_j}$ for window $j$, where $N_j$ is the number of segments.

**Temporal modeling via Transformer.** We use a Transformer encoder to model dependencies across movement segments. Given the sequence $\{z_j^n\}_{n=1}^{N_j}$, a 5-layer Transformer encoder produces contextual embeddings $\{u_j^n\}_{n=1}^{N_j}$. Because segment timing is irregular, standard absolute/sinusoidal positional encodings—which implicitly assume roughly uniform spacing—are unsuitable. We therefore incorporate explicit time information through time-aware positional encodings.

Let $t_j^n$ denote the midpoint time of token $n$ within window $j$ (stored as metadata), and let $\tilde{t}_j^n \in [0, 1]$ be its normalized time within the 10 s window. We encode absolute phase by adding an MLP time embedding $p(\tilde{t}_j^n)$:

$$\tilde{z}_j^n = \text{LN}\left(z_j^n + p(\tilde{t}_j^n)\right). \tag{3}$$

For relative timing, we compute pairwise offsets $\Delta t_j^{i,k} = t_j^i - t_j^k$ and normalize by the window's median inter-token interval. Intuitively, this approximates the number of typical inter-segment "steps" between segments $i$ and $k$. We discretize the normalized offsets, index a learned relative-position embedding, and add the resulting bias to the attention logits. We include an ablation of positional information in Appendix F.1.

**Masked movement segment reconstruction.** We pretrain the model by reconstructing masked movement segments from surrounding context. To encourage both local interpolation and long-range inference, we use a hybrid masking policy. For each window, we sample one of two masking schemes with equal probability: (i) random masking, which masks a fraction $r$ of tokens, and (ii) contiguous time masking, which partitions the 10 s window into 1 s bins, samples a fraction $r$ of these bins, and masks any segment overlapping with them. After CNN encoding, masked tokens are replaced with a learned `[MASK]` embedding. We set the masking rate to $r=0.5$, selected empirically by sweeping $r \in \{0.25, 0.50, 0.75\}$ (see Appendix Table 3).

To discourage trivial copying from visible context, we additionally corrupt visible tokens. For $20\%$ of unmasked tokens, we replace the CNN embeddings $h$ with one randomly sampled from another token in the same window, while leaving the original metadata (axis, duration, time) intact. A decoder then reconstructs the resampled waveform $\bar{\mathcal{S}}_j^n$ for each valid token. We minimize an $\ell_1$ reconstruction loss over non-padding tokens, upweighting masked tokens by a factor of 100.

## 3.5. Downstream Embeddings

Given a window $\mathcal{X}_j$, the Transformer produces contextual token embeddings $\{u_j^n\}_{n=1}^{N_j}$ for the movement segments in that window. We summarize these embeddings using mean and standard-deviation pooling and concatenate them to form a window-level segment representation $T_j \in \mathbb{R}^{128}$.

**Re-incorporating gravity.** Because our tokenization operates on gravity-reduced acceleration, $T_j$ lacks information regarding device orientation and rotation. To address this during downstream linear probing, we re-incorporate the low-frequency residual $g_j$ by low-pass filtering the raw acceleration stream using the matched cutoff of 0.5 Hz. We resample $g_j$ to 300 samples per axis and flatten it into $\text{flat}(g_j) \in \mathbb{R}^{900}$. The fused probe feature is thus $F_j = [T_j, \text{flat}(g_j)] \in \mathbb{R}^{1028}$. This fusion simply restores the gravitational context discarded during tokenization, maintaining a fair comparison against baselines that process raw acceleration.

## 3.6. Comparative Baselines and Reference Models

To ensure a fair comparison and isolate the impact of our proposed tokenization, we pretrain all baselines on the same NHANES corpus with matched windowing and sampling, and evaluate all methods under the same subject-disjoint transfer protocol. We compare our approach against two distinct categories of baselines.

First, we consider three representative wearable SSL configurations: (i) contrastive learning (TF-C) (Zhang et al., 2022), (ii) augmentation prediction (AugPred; replicating the architecture and objective of Yuan et al. (2024)), and (iii) a naive tokenization baseline that performs masked reconstruction over equal-length chunks. The equal-chunking baseline serves as a direct ablation of our tokenization strategy: it keeps the masked-reconstruction objective fixed while replacing biologically meaningful movement-segment tokens with arbitrary time chunks of the raw signal. Upstream training metrics and comprehensive implementation details (e.g., optimizers, model sizes, and baseline architectures) are provided in Appendices C and D.

Second, to provide a broader context, we include two general-purpose time-series foundation models: Chronos (Ansari et al., 2024) and Moment (Goswami et al., 2024). These models are tested using their publicly released checkpoints without any domain-specific pretraining.

## 3.7. Evaluation: Human Activity Recognition

To ensure cross-subject generalization, we employ *subject-disjoint* data splits across all experiments. For datasets with $\leq 10$ subjects, we use leave-one-subject-out cross-validation (LOSOCV). For datasets with $> 10$ subjects, we use five-fold subject-wise cross-validation. In all cases, we report the Macro-F1 on the held-out test subjects, expressed as mean $\pm$ standard deviation across the respective splits.

**Linear probing protocol.** We adopt linear probing as our

Table 1. Macro-F1 (mean ± std) on six subject-disjoint HAR benchmarks using frozen representations and multinomial logistic regression. All wearable SSL baselines are pretrained on the same NHANES upstream corpus under a matched transfer protocol; generic time-series foundation models are included only as contextual reference and are not pretrained on NHANES.

| Model | UMH | PAMAP | WISDM | MHEALTH | WHARF | HAD | Avg. |
|---|---|---|---|---|---|---|---|
| *Generic Time Series FMs* | | | | | | | |
| Chronos (8.3M) | 0.23 ± 0.02 | 0.56 ± 0.04 | 0.53 ± 0.02 | 0.63 ± 0.05 | 0.25 ± 0.04 | 0.34 ± 0.07 | 0.42 (n=6) |
| Moment (35M) | 0.48 ± 0.04 | 0.64 ± 0.07 | 0.68 ± 0.03 | 0.66 ± 0.06 | 0.37 ± 0.14 | 0.72 ± 0.04 | 0.59 (n=6) |
| *Controlled (NHANES pretrained)* | | | | | | | |
| Mask-Recon (1.4M) | 0.39 ± 0.04 | 0.61 ± 0.05 | 0.58 ± 0.02 | 0.62 ± 0.06 | 0.30 ± 0.09 | 0.33 ± 0.07 | 0.47 (n=6) |
| AugPred (10.5M) | 0.40 ± 0.03 | 0.59 ± 0.07 | 0.60 ± 0.02 | 0.67 ± 0.08 | 0.31 ± 0.07 | 0.69 ± 0.03 | 0.54 (n=6) |
| Contrastive (TF-C) (8.7M) | 0.53 ± 0.07 | 0.62 ± 0.13 | 0.64 ± 0.04 | 0.68 ± 0.17 | 0.37 ± 0.12 | 0.72 ± 0.08 | 0.59 (n=6) |
| **Bio-PM** (1.4M) | **0.57 ± 0.05** | **0.69 ± 0.16** | **0.70 ± 0.03** | **0.80 ± 0.08** | **0.41 ± 0.03** | **0.75 ± 0.06** | 0.65 (n=6) |

evaluation protocol to ensure architectural fairness, as it directly measures what the SSL pretraining encodes without any task-specific adaptation. By freezing the encoder backbone, we evaluate the inherent quality of the pretrained representations rather than a model's capacity to compensate for weak features during supervised fine-tuning. This isolation is particularly crucial given that the baseline architectures vary significantly in size, ranging from 1.4M to 10.5M parameters. To further guarantee a fair assessment of tokenization, we pretrained all SSL baselines on the same large-scale NHANES corpus rather than relying on off-the-shelf checkpoints.

For all 10 s windows, we extract feature representations using the frozen pretrained encoder. Then, for every split, we (i) fit a z-score normalization on the training subjects and apply it to the train and test sets, (ii) select $C$ by cross-validating on the training subjects to maximize the Macro-F1, (iii) train a multinomial logistic regression using the optimal $C$ on the full training set, and (iv) evaluate the Macro-F1 on held-out test set. To contextualize the probe capacity, we report the number of trainable parameters for each representation variant in Appendix Table 6.

## 4. Results

Under a controlled upstream setting where all SSL baselines are pretrained on the same NHANES corpus and evaluated with the same subject-disjoint linear-probe protocol, we find:

(i) Bio-PM transfers best across all six HAR benchmarks; (ii) movement segment tokenization is a major contributor to the gains, as replacing it with structure-agnostic equal chunking degrades transfer by 0.18 Macro-F1 on average; and (iii) Bio-PM is more label-efficient and encodes the sequential structure of human movements.

### 4.1. Comparing SSL Objectives

**Evaluating Transfer via Linear Probe** Table 1 presents the Macro-F1 scores for frozen linear probes across the six HAR benchmarks. Bio-PM strictly dominates the compared baselines, achieving the top scores on every dataset and the highest overall average (0.65 Macro-F1). When compared to the strongest baseline, TF-C, Bio-PM provides consistent improvements across the board (average +0.06), with the most significant margin occurring on MHEALTH (0.80 vs. 0.68, a +0.12 improvement). Similarly, Bio-PM demonstrates substantial average improvements of +0.11 and +0.18 over AugPred (Yuan et al. (2024)'s methodology) and the capacity-matched masked-reconstruction baseline, respectively. To statistically test Bio-PM's gains over TF-C, we performed a one-sided Wilcoxon signed-rank test over 41 folds pooled across the six datasets. Bio-PM outperformed on 31/41 folds ($p < 0.0001$, Cohen's $d = 0.61$); per-dataset tests were mixed due to limited fold counts.

*Table 2.* Macro-F1 (mean ± std. dev.) under controlled ablations isolating key design choices: gravity, tokenization, and pretraining.

| Model | UMH | PAMAP | WISDM | MHEALTH | WHARF | HAD | Avg. |
|---|---|---|---|---|---|---|---|
| Bio-PM | 0.57 ± 0.05 | 0.69 ± 0.16 | 0.70 ± 0.03 | 0.80 ± 0.08 | 0.41 ± 0.03 | 0.75 ± 0.06 | 0.65 ± 0.07 |
| Bio-PM w/o Gravity | 0.48 ± 0.04 | 0.60 ± 0.04 | 0.67 ± 0.02 | 0.70 ± 0.04 | 0.33 ± 0.08 | 0.55 ± 0.05 | 0.56 ± 0.05 |
| Bio-PM w. Naive Tokenization | 0.39 ± 0.04 | 0.61 ± 0.05 | 0.58 ± 0.02 | 0.62 ± 0.06 | 0.30 ± 0.09 | 0.33 ± 0.07 | 0.47 ± 0.06 |
| Bio-PM w/o Pretraining | 0.45 ± 0.13 | 0.59 ± 0.14 | 0.58 ± 0.02 | 0.67 ± 0.12 | 0.41 ± 0.11 | 0.61 ± 0.09 | 0.55 ± 0.10 |

## 4.2. Ablation Studies

**Isolating tokenization strategy.** We isolate the effect of our tokenization strategy by replacing our movement-segment tokens with uniform, equal-length raw-signal chunks while holding all other pretraining and evaluation factors constant, including pretraining objective, model capacity, optimizer, masking rate, and downstream linear-probing protocol. To guarantee that this ablation tests the tokenization strategy rather than the temporal resolution, we set the chunk duration to match the average length of a movement segment ($\approx 0.5$ s at 80 Hz). We then resample each chunk to 32 samples before processing it using the exact same CNN encoder as Bio-PM.

Replacing our biologically grounded segments with uniform chunks causes the average Macro-F1 score to drop from 0.65 to 0.47 (-0.18), with the most severe degradations observed on the HAD, UMH, and MHEALTH datasets (Table 2). Because all other architectural and optimization variables are identical, this performance gap supports that our biologically motivated tokenization provides a highly effective, task-relevant inductive bias for transfer learning.

**Role of the low-frequency gravity residual.** Our tokenization operates on gravity-attenuated acceleration to emphasize voluntary motion. However, the residual low-frequency component contains rotation and orientation cues that are often critical for downstream tasks. As anticipated, providing this component to the downstream linear probe leads to a substantial improvement, increasing the average Macro-F1 from 0.56 to 0.65, with the largest gain of +0.20 on HAD (Table 2; also see Figures 14-18 in Appendix H).

Notably, Bio-PM remains highly competitive even without the low-frequency data (0.56 Macro-F1 on average), continuing to exceed both the uniform-chunk masked-reconstruction baseline (0.47) and AugPred (0.54) (Table 2). This further supports that movement-aligned tokenization acts as a strong, standalone inductive bias.

**Role of self-supervised pretraining.** Fine-tuning the identical architecture end-to-end without SSL pretraining reduces the average Macro-F1 from 0.65 to 0.55. (Table 2). End-to-end fine-tuning updates the entire 1.4M-parameter encoder, whereas the linear probing protocol trains only a lightweight classifier head ($\sim$8k–26k parameters, depending on the dataset). Crucially, the benefits of pretraining manifest in two ways: higher mean performance (e.g., UMH, WISDM, HAD) and improved reliability across splits (e.g., WHARF exhibits similar mean but substantially reduced variance after pretraining). Together, these results support that Bio-PM encodes robust, transferable features that are difficult to learn from limited labeled data alone.

## 4.3. Data efficiency in low-subject regimes

In wearable HAR, recruiting diverse participants and annotating their data is highly resource-intensive. We therefore evaluate data efficiency by scaling down the number of labeled training subjects, which simultaneously restricts subject diversity and reduces labeled data volume (held-out test subjects remain fixed).

As shown in Figure 3, Bio-PM consistently matches or exceeds baseline performance across all subject budgets, yielding particularly clear improvements on the two largest benchmarks (MHEALTH and WISDM). Importantly, the relative baseline rankings from the full-data setting persist in these low-subject regimes, indicating that our tokenization remains highly effective even when labeled data is scarce.

## 4.4. Generalization to Unseen Transitions

Complex human behaviors are often compositional: they reuse a finite set of recurring motion units in different orders to produce diverse activities. We therefore test whether Bio-PM understands how motion units function in context by evaluating next-token prediction on *unseen* transitions.

**Experimental setup.** For each dataset, we first construct a discrete vocabulary by clustering all movement-segment token embeddings via $K$-means. We sweep $K \in$

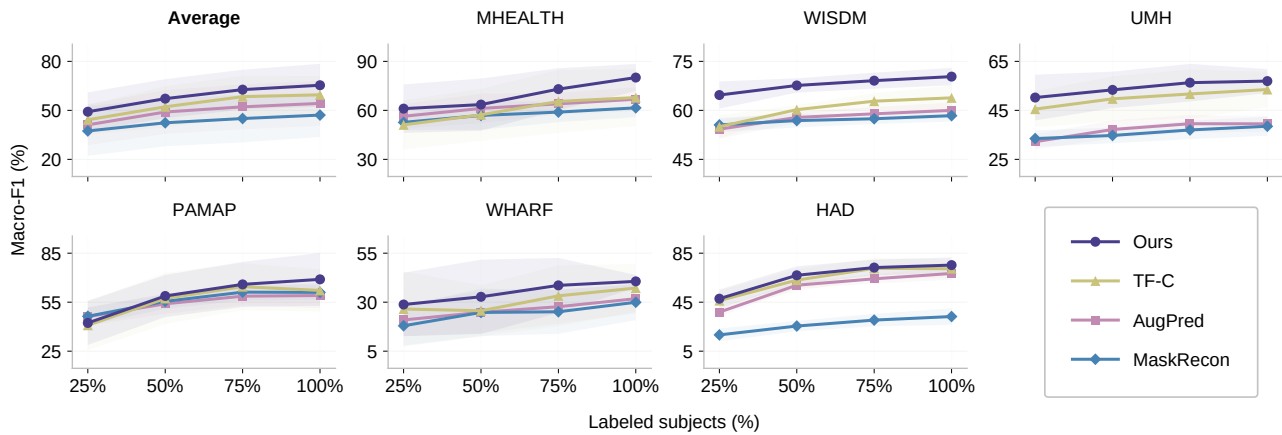

*Figure 3.* Downstream Macro-F1 across varying labeled-subject budgets. Bio-PM maintains strong performance in low-resource regimes and scales more effectively than controlled SSL baselines as training data increases.

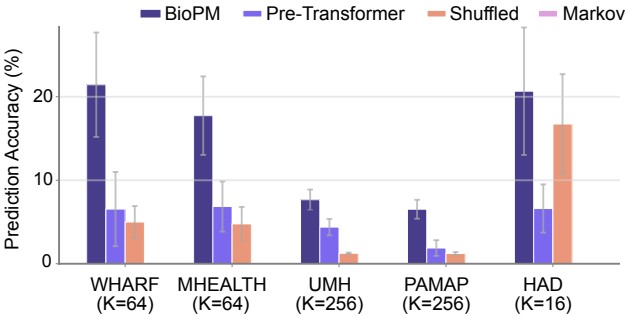

*Figure 4.* Next-token prediction accuracy on unseen token transitions using Bio-PM embeddings. HAD instability reflects its much smaller transition set ($\approx$250 vs. 2.3k–27k).

$\{16, 32, 64, 128, 256\}$ and select the optimal $K$ based on the highest silhouette score (Figure 4). This process converts each 10 s window into a sequence of discrete cluster IDs, $(p_1, \ldots, p_N)$, where $p_i \in [1, K]$ and $N$ represents the number of movement segments within the window. We then train a linear classifier to predict the next discrete ID $p_{i+1}$ given the Bio-PM embedding of the current token $p_i$ as input. Most importantly, we evaluate this probe using a held-out bigram split: specific transitions (e.g., $(A \rightarrow B)$) present in the test set are explicitly excluded from the training set. This forces the model to generalize based on learned structural rules rather than memorized bigram frequencies.

**Baselines.** To evaluate the importance of temporal context, we compare Bio-PM's post-Transformer embeddings against: (i) non-contextual embeddings (pre-Transformer CNN features from Equation (2)), to test the value of temporal modeling; (ii) a Markov baseline that predicts the most frequent successor given $p_i$ observed during training; and (iii) a Shuffle control, where tokens within a window are randomly permuted to destroy temporal ordering during both training and testing.

**Results.** As shown in Figure 4, Bio-PM's contextual embeddings consistently outperform the non-contextual baselines on unseen transitions across all datasets, yielding improvements of $+3.3$ to $+14.9$ percentage points. The results demonstrate that Bio-PM learns the underlying sequential structure of the data rather than just local, token-level cues. By internalizing these high-level patterns, the model successfully generalizes its predictive capabilities to entirely unseen pairings of movement segments.

### 4.5. Qualitative Analysis of Activity Confusions

To understand how encoding the compositional structure of human movement contributes to HAR, we qualitatively compare Bio-PM against our strongest controlled baseline, TF-C (Appendix H). Two prominent patterns are observed. First, Bio-PM yields substantial improvements on activities defined by the temporal arrangement of their sub-activities: transitional activities like stand-to-sit/sit-to-stand (WHARF: 0.50 vs. 0.21), stand-to-lying/lying-to-stand (WHARF: 0.17 vs. 0.02), along with compositional activities like cleaning (UMH: 0.80 vs 0.67). Second, Bio-PM more effectively distinguishes activities with similar inertial signatures, such as stair climbing vs. flat walking (WISDM: 0.81 vs 0.54; WHARF: 0.34 vs 0.09), tennis serve vs. right-hand throw (HAD: 0.78 vs 0.44). The first pattern reflects the Bio-PM's sensitivity to segment order; the second, to segment identity.

### 4.6. Robustness to Practical Perturbations

We analyze the robustness of Bio-PM to two practical deployment concerns: (i) additive sensor noise and (ii) the high-pass cutoff frequency used for gravity separation.

**Noise robustness.** Sensor noise can degrade tokenization quality by introducing spurious zero-crossings. While our two-stage hysteresis filter could merge such noise-induced

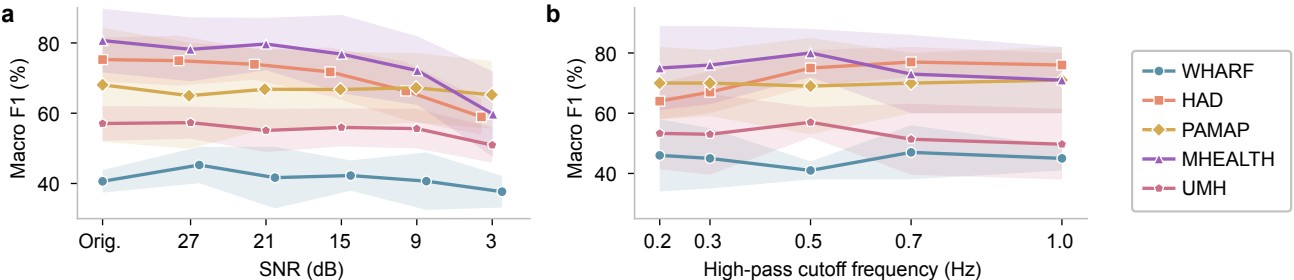

*Figure 5.* Robustness analyses of Bio-PM against (a) additive sensor noise and (b) high-pass cutoff frequency for gravity separation.

tokens (Section 3.4; Appendix F.2), we empirically test the system's robustness by injecting additive white Gaussian noise (AWGN) at test time. The noise mean and variance are matched to empirical accelerometer measurements (Nirmal et al., 2016). We simulate noise floors 1.5x, 2x, 3x, 5x, and 9x above typical sensor levels, corresponding to signal-to-noise ratios (SNRs) of $\approx$ 27, 21, 15, 9, and 3 dB, respectively. As shown in Figure 5a, the Macro-F1 remains broadly stable down to 15 dB (3x typical noise) and degrades substantially only at 3 dB.

**High-pass cutoff sensitivity.** Our tokenization isolates voluntary motion from gravity using a 0.5 Hz high-pass filter (Section 3.3). To test sensitivity to this choice, we sweep the cutoff across $\{0.2, 0.3, 0.5, 0.7, 1.0\}$ Hz during linear probing, keeping the pretrained model fixed. As shown in Figure 5b, the Macro-F1 varies by only a few points across this range with no consistent directional trend, indicating that the learned representations are robust to the subtle gravity-separation threshold.

## 5. Discussion

Our evaluation demonstrates that Bio-PM significantly outperforms existing baselines across six diverse HAR benchmarks, driven primarily by the strong inductive bias of our bio-inspired tokenization strategy. Furthermore, the model exhibits improved label efficiency, robust stability against sensor noise, and the ability to accurately predict unseen kinematic transitions. Together, these findings support the conclusion that Bio-PM successfully encodes the compositional structure of human movements.

Although the present work validates movement-segment tokenization specifically on wrist-worn accelerometer data, the underlying submovement theory (Hogan & Sternad, 2012) applies broadly to the linear and angular kinematics of various body joints. This suggests that our tokenization framework—and by extension, the pretrained model—could adapt naturally to gyroscope and magnetometer data, where angular velocity can similarly be described as a superposition of bell-shaped primitives. Furthermore, although we focused on wrist-worn sensing due to its ubiquity in

long-term monitoring, prior evidence indicates that submovement theory also governs other end-effectors, including lower-limb locomotion (Hogan & Sternad, 2013), head movements (Chen et al., 2012), and discrete ankle movements (Michmizos et al., 2014). Extending Bio-PM to these diverse sensing modalities and body locations remains an important direction for future work.

Beyond standard HAR, the proposed tokenization strategy and pretrained model hold significant promise for clinical research, particularly for assessing motor severity in neurological conditions using continuous, naturalistic wrist accelerometry. Given that submovement-based features have been extensively demonstrated to be clinically relevant for evaluating patients with stroke (Wang et al., 2025), Parkinson's disease (Noy et al., 2025), and Huntington's disease (Nunes et al., 2025), adapting Bio-PM for automated clinical motor assessments represents an exciting next step.

This study has several limitations. First, the proposed pretraining framework completely excludes the low-frequency gravitational components of the accelerometer data, meaning postural and rotational context is not incorporated during pretraining. Incorporating these cues jointly with movement-segment tokens remains an important direction for future work. Second, although NHANES provides large-scale population coverage, the learned movement statistics may still reflect demographic and cultural biases specific to the dataset. Third, while the model benefits from pretraining on approximately 11,000 participants, scaling to substantially larger wearable corpora—such as the UK Biobank, which contains unlabeled data from over 100,000 individuals—may further improve downstream generalization. Fourth, the downstream benchmarks consist primarily of scripted or semi-natural activities, so generalization to fully unstructured free-living behavior remains untested. Ultimately, addressing these data and architectural limitations will help transition this biologically grounded framework into a truly universal foundation model for human movements.

## Impact Statement

This paper advances self-supervised representation learning for wrist-worn accelerometer data to improve data efficiency in human activity recognition. Our experiments use publicly available, de-identified datasets and are intended for research use on activity-recognition benchmarks. Because movement patterns can contain identifying behavioral signatures, privacy and re-identification risks should be carefully considered in downstream deployment settings. Any real-world deployment would require context-specific validation and adherence to applicable data governance and privacy practices.

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

# A. Tokenization Pipeline

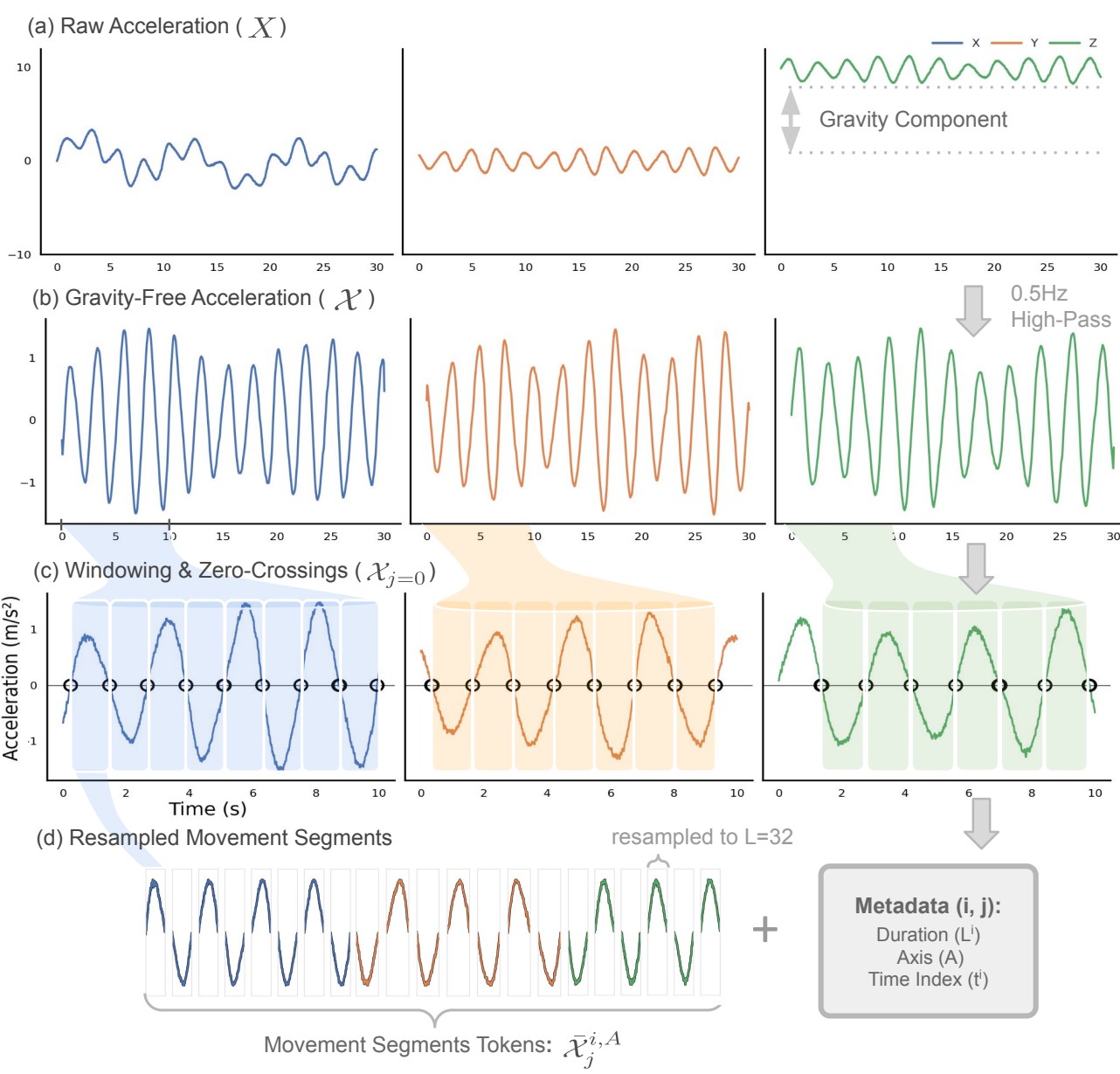

*Figure 6.* End-to-end conversion from raw wrist accelerometer signals to fixed-length movement-segment tokens: gravity separation via filtering, zero-crossing boundary detection, segmentation, and resampling to a common token length, along with token metadata used for sequence modeling.

# B. Our Model: Upstream Analysis

Ablations in Tables 3 and 4 are reported on a subset of benchmarks (PAMAP, MHealth, WHARF, HAD). WISDM was excluded due to the substantial computational cost of repeated sweeps on its 51-subject cross-validation, and UMH was added to our benchmark suite after these ablations were completed.

### B.0.1. VARYING UPSTREAM MASKING RATE

See table 3. We chose the model pretrained on a masking rate of 50%.

*Table 3.* **Effect of masking rate on linear-probe Macro-F1 for Bio-PM**

| Masking Rate | PAMAP | MHEALTH | WHARF | HAD | Avg |
|---|---|---|---|---|---|
| 25% | 0.69 ± 0.12 | 0.80 ± 0.08 | 0.40 ± 0.02 | 0.71 ± 0.06 | 0.65 (n=4) |
| 50% | **0.71** ± **0.12** | **0.80** ± **0.08** | 0.42 ± 0.03 | **0.73** ± **0.06** | 0.67 (n=4) |
| 75% | 0.69 ± 0.13 | 0.79 ± 0.10 | 0.41 ± 0.05 | 0.71 ± 0.07 | 0.65 (n=4) |

### B.0.2. VARYING MODEL SIZE.

We evaluated whether increasing Bio-PM capacity improves transfer under our largest upstream setting (11k subjects). Across downstream HAR benchmarks, scaling from 1.4M to 6M and 13M parameters yields small consistent gains in linear-probe Macro-F1 (Table 4). However, we use the 1.4M model throughout for three practical reasons: (i) it is the only configuration whose embedding dimension is closest to the Yuan et al.'s ResNet baseline (1.4M: 1028-d; 6M: 1156-d; 13M: 1412-d; ResNet: 1024-d), avoiding improvements attributable to a higher-dimensional probe space; (ii) HAR is commonly deployed on-device, making smaller backbones more suitable for edge inference; and (iii) it substantially reduced training cost and enabled broader ablations internally, while still outperforming prior methods in our main comparisons.

Concretely, the 1.4M model uses width $D=64$ with 5 Transformer layers, while the 6M and 13M variants use $D=128$ and $D=256$ respectively, each with 10 layers.

*Table 4.* **Effect of model size on linear-probe Macro-F1 for Bio-PM**

| Model Size | PAMAP | MHEALTH | WHARF | HAD | Avg |
|---|---|---|---|---|---|
| 1.4M | 0.71 ± 0.12 | 0.80 ± 0.08 | 0.42 ± 0.03 | 0.73 ± 0.06 | 0.67 (n=4) |
| 6M | 0.69 ± 0.13 | 0.82 ± 0.1 | 0.43 ± 0.05 | 0.71 ± 0.08 | 0.66 (n=4) |
| 13M | 0.69 ± 0.13 | 0.82 ± 0.09 | 0.44 ± 0.05 | 0.76 ± 0.04 | 0.68 (n=4) |

### B.0.3. UPSTREAM DATA SAMPLING

To efficiently sample informative segments from long, free-living NHANES wrist accelerometer recordings, we use an activity-index–driven window selection strategy inspired by the Activity Index (AI) of Bai *et al.* (Bai et al., 2016). Continuous streams are segmented into non-overlapping 10 s windows, and each window is assigned a scalar activity score following the AI formulation. Windows with low activity (AI < 50) are discarded to remove near-stationary behavior that dominates free-living data. The remaining windows are stratified by activity intensity: household and moderate-intensity activities are subsampled due to their high prevalence, while all high-intensity activity windows are retained given their relative scarcity. This yields a compact yet diverse training set spanning a range of movement intensities.

### B.0.4. UPSTREAM: RECONSTRUCTION PERFORMANCE

For upstream metrics by masking rate, see Figure 7. For examples of reconstruction, see Figure 8.

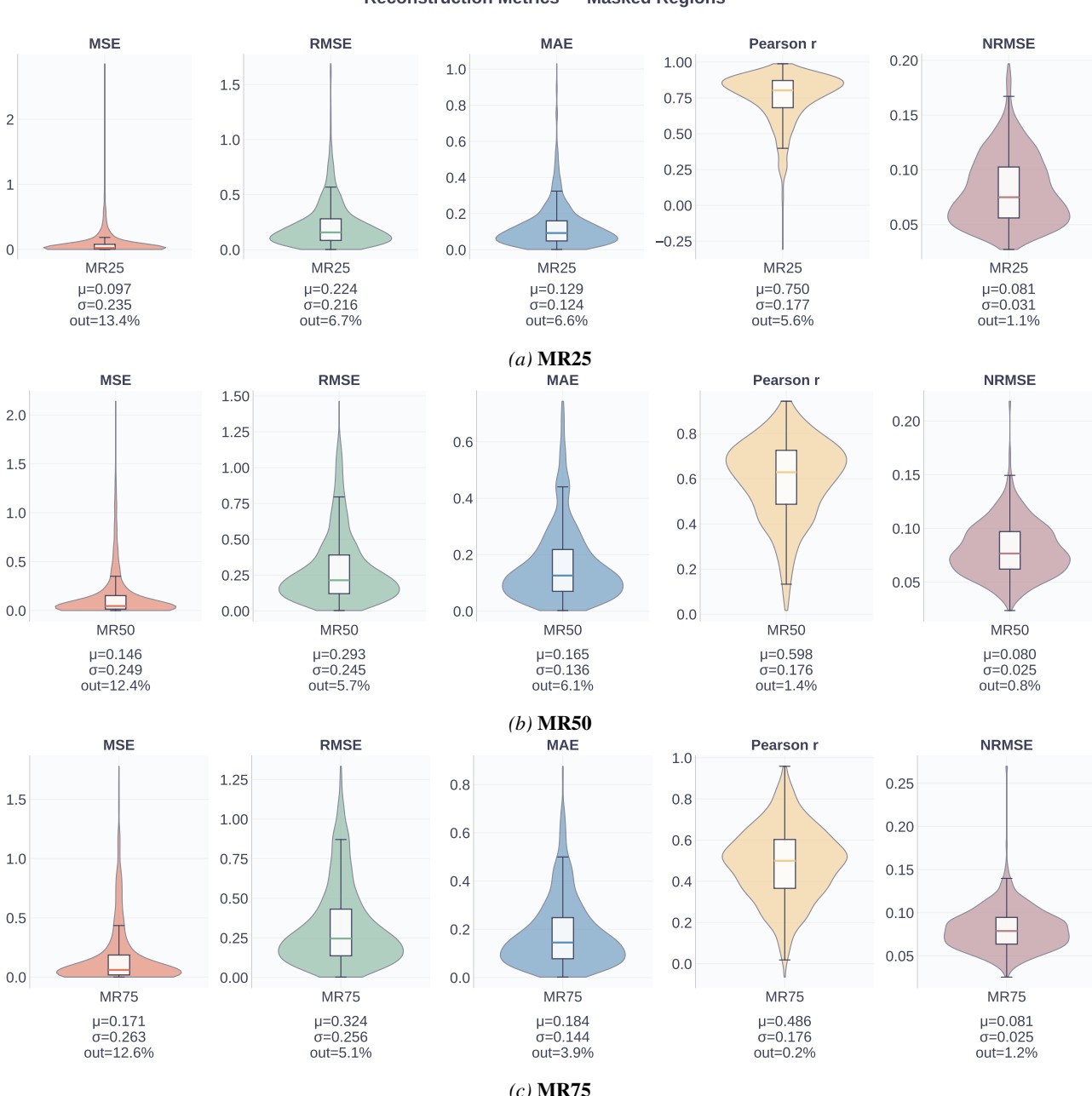

*Figure 7.* **Upstream masked-region reconstruction quality under different masking rates.** Violin plots show the distribution of reconstruction performance evaluated only on masked regions for MR25/50/75 using MSE, RMSE, MAE, Pearson $r$, and NRMSE. Boxes indicate median and interquartile range (whiskers: $1.5\times$IQR). Numbers below each subplot report mean $\mu$, standard deviation $\sigma$, and outlier fraction. Increasing masking increases error (MSE $0.085 \rightarrow 0.145$, MAE $0.128 \rightarrow 0.177$) and reduces correlation ($r = 0.760 \rightarrow 0.518$), while NRMSE stays roughly constant ($\approx 0.081$–$0.083$).

## C. Baseline Upstream Performances

**AugPred.** Pretext accuracies: permutation 89.5%, time reversal 89.2%, and time warp 95.8% (comparable to the converged values reported by Yuan et al. (2024)).

**TF-C.** Validation contrastive losses: $L_{tt}$=5.4298, $L_{ff}$=5.4208, $L_{tf}$ =4.8183 (total 15.6689).

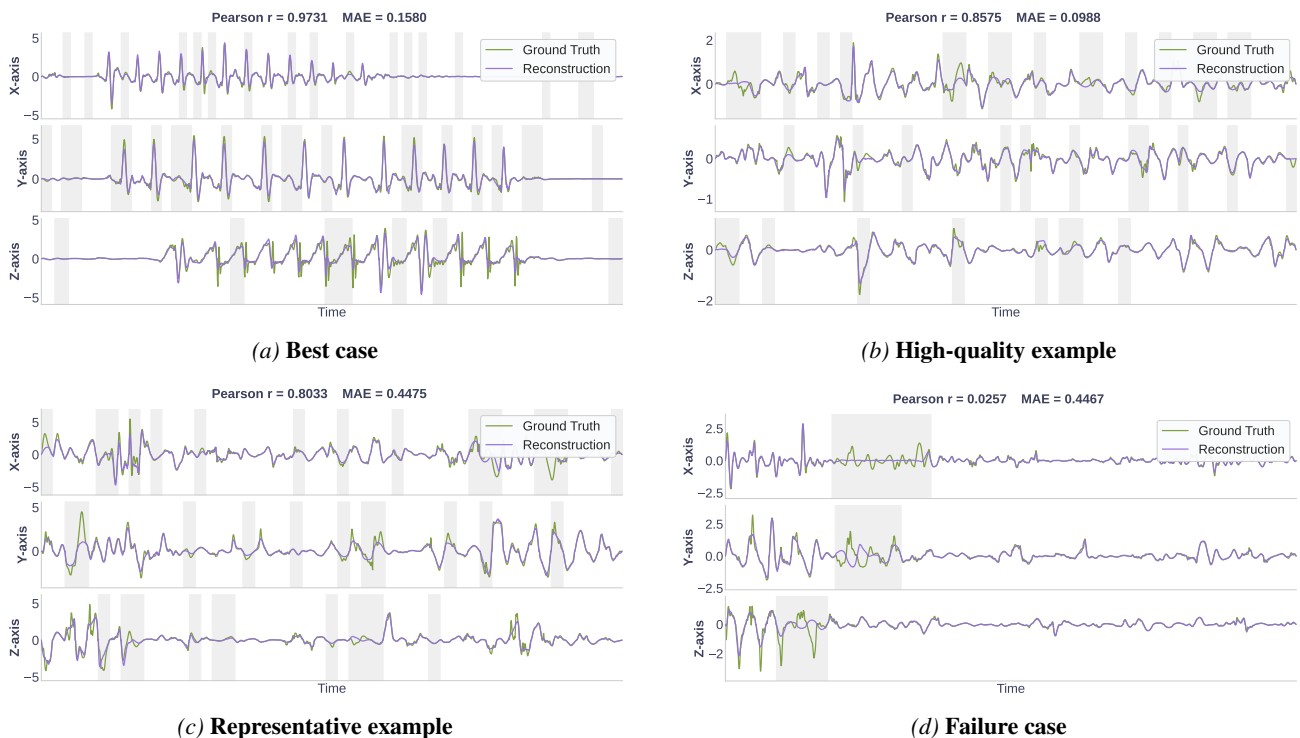

*Figure 8.* **Qualitative masked-region reconstruction examples.** Each panel visualizes the masked-movement-segment pretraining task on a representative 1D channel: the *top* subplot shows the input with masked spans removed (gray bands denote masked regions), and the *bottom* subplot overlays the ground-truth signal with the model reconstruction. Pearson $r$ (shown in titles where available) summarizes agreement between reconstruction and ground truth on masked movement segments. These examples illustrate the range of outcomes from near-perfect recovery (e.g., $r \approx 0.99$) to clear failure modes ($r \approx 0.12$), where the model under-reconstructs sharp transients and high-amplitude bursts within masked segments.

**Masked reconstruction (equal-chunk).**   Masked-token MAE: 0.35.

For each baseline, we select the checkpoint that maximizes validation Macro-F1 under the linear-probing protocol. Additional training for TF-C led to degraded validation upstream performance.

# D. Implementation Details

**Training setup.**   Unless otherwise stated, all models are implemented in PyTorch. We pretrain all SSL methods on the same NHANES corpus with matched windowing and sampling.

**Masked reconstruction baselines (capacity-matched).**   For masked reconstruction, our method and the equal-chunk baseline use the same Transformer-based architecture and identical optimization settings (Adam, learning rate $10^{-4}$, batch size 512). This isolates the effect of tokenization while holding objective, optimizer, and model capacity fixed.

**AugPred replication (Yuan et al.).**   For AugPred, we replicate the objective and augmentation pipeline of Yuan et al. (2024) and use the same backbone architecture as in their work: an 18-layer ResNet-V2 with 1D convolutions (approximately 10.5M parameters), producing a 1024-dimensional representation.

We follow Yuan et al. (2024) with two small implementation adjustments for our matched-upstream setting—retaining 80 Hz (no 40 Hz downsampling) and omitting weighted sampling due to our existing upstream filtering (see Sec. B.0.3). Nevertheless, we get comparable pretext accuracies as reported by Yuan et al. (2024) (Appendix C).

**TF-C architecture and parameter budgeting.**   TF-C uses two encoders (temporal and frequency branches). To keep the total parameter budget comparable to the single-encoder baselines, we reduce the per-encoder width such that the combined TF-C model has approximately 8.7M parameters.

**Bio-PM model size.** Our default Bio-PM model has approximately 1.4M parameters. We also experimented with larger variants (e.g., 6M and 13M parameters) and observed similar downstream trends; we report the 1.4M model to prioritize efficiency and practical deployment considerations.

**Computational cost of tokenization.** On a single CPU core, tokenizing a batch of 32 MHealth windows takes $\approx 0.4$ s with our movement-segment pipeline versus $\approx 0.15$ s for naive equal-length chunking — a $\approx 2.7$x overhead, modest relative to the downstream transfer gains.

## E. Datasets and Preprocessing

*Table 5.* Summary of downstream HAR datasets.

| Dataset | Setting | #Sub | #Classes | Hz | References |
|---|---|---|---|---|---|
| NHANES Accelerometer | Free-living | $\approx$11k | – | 80 | (Centers for Disease Control and Prevention (CDC), 2010) |
| UMH | Semi-natural | 25 | 29 | 100 | (Casilari et al., 2024) |
| PAMAP | Lab protocol | 8 | 8 | 100 | (Reiss, 2012) |
| WISDM (Actitracker) | Semi-natural | 51 | 18 | 20 | (Weiss, 2019) |
| MHEALTH | Scripted protocol | 10 | 11 | 50 | (Banos & Saez, 2014; Banos et al., 2014; 2015) |
| WHARF | In-home scripted | 17 | 14 | 32 | (Bruno et al., 2014) |
| HAD | Lab multimodal | 8 | 27 | 50 | (Chen et al., 2015) |

**Wrist-only sensor.** For multi-sensor datasets, we retain only the wrist/hand accelerometer stream and discard other body locations/modalities.

**Filtering** We compute (i) voluntary acceleration using a 6th-order Butterworth highpass filter (0.5Hz), and (ii) gravity using a 6th-order Butterworth low-pass filter (cutoff 0.5 Hz).

**Unit normalization.** All datasets were converted from m/s$^2$ to $g$ (divide by 9.80665).

**Labeling.** For each 10 s window, labels are assigned by majority vote within each window; windows dominated by transition/unknown labels are discarded.

**Resampling.** To ensure a consistent temporal resolution across datasets, all accelerometer signals are resampled to a common sampling rate using interpolation-based resampling. This step standardizes window lengths and frequency content across subjects and recording setups.

**Windowing.** The preprocessed signals are segmented into fixed-length sliding windows of 10 s with overlap. Windowing enables learning from short, locally stationary motion segments while increasing the effective number of training samples.

**Quality control.** Windows containing insufficient valid samples or dominated by null, transition, or excluded activity labels are removed. This filtering step reduces label noise and ensures that retained windows represent coherent activities.

**Motion magnitude normalization.** For each window, motion intensity is characterized using the mean absolute deviation (MAD) of the filtered acceleration magnitude. This statistic provides a robust, scale-invariant measure of movement variability and is used for downstream analysis and dataset characterization.

## F. Miscellaneous

### F.1. Sanity Checks and Ablations

Experiments conducted on all datasets except WISDM because of its size.

**Does the transformer use temporal information?** As a sanity check, we ablate positional information *at transfer time* by disabling positional encodings in the frozen pretrained encoder and retraining only the linear probe. Removing positional encodings reduces Macro-F1 by $\approx$**3 points** on average, with the largest drop on HAD (**8.2** points). This suggests that the learned representations exploit token timing in addition to token content, particularly for order-dependent activities.

### F.2. Zero-crossing hysteresis for movement-segment extraction

We extract candidate segment boundaries as sign changes in the 1D gravity-free acceleration signal. To reduce noise-induced crossings when the signal hovers near zero, we apply two heuristics:

**Temporal hysteresis.** Enforce a minimum inter-crossing interval of 50 ms. Candidate crossings within 50 ms of the last accepted crossing are discarded.

**Amplitude hysteresis.** For each provisional segment $\mathcal{S}$ between consecutive accepted crossings, compute peak amplitude $\max_{t \in \mathcal{S}} |x(t)|$. If two consecutive segments both have peak amplitude below 0.01, we remove their intermediate crossing and merge the two segments.

After hysteresis, each segment is resampled to a fixed length $L{=}32$ via linear interpolation.

## G. Linear Probing Parameter Comparison

*Table 6.* Number of learned parameters in linear probing. Only logistic regression weights are trained; encoder weights remain frozen.

| Dataset | Classes ($K$) | AugPred (1024-d) | TF-C (512-d) | Transformer (128-d) | Bio-PM (1028-d) |
|---|---|---|---|---|---|
| UMH | 29 | 29,725 | 14,877 | 3,741 | 29,841 |
| PAMAP | 8 | 8,200 | 4,104 | 1,032 | 8,232 |
| MHEALTH | 11 | 11,275 | 5,643 | 1,419 | 11,319 |
| HAD | 27 | 27,675 | 13,851 | 3,483 | 27,783 |

## H. Confusion Matrices

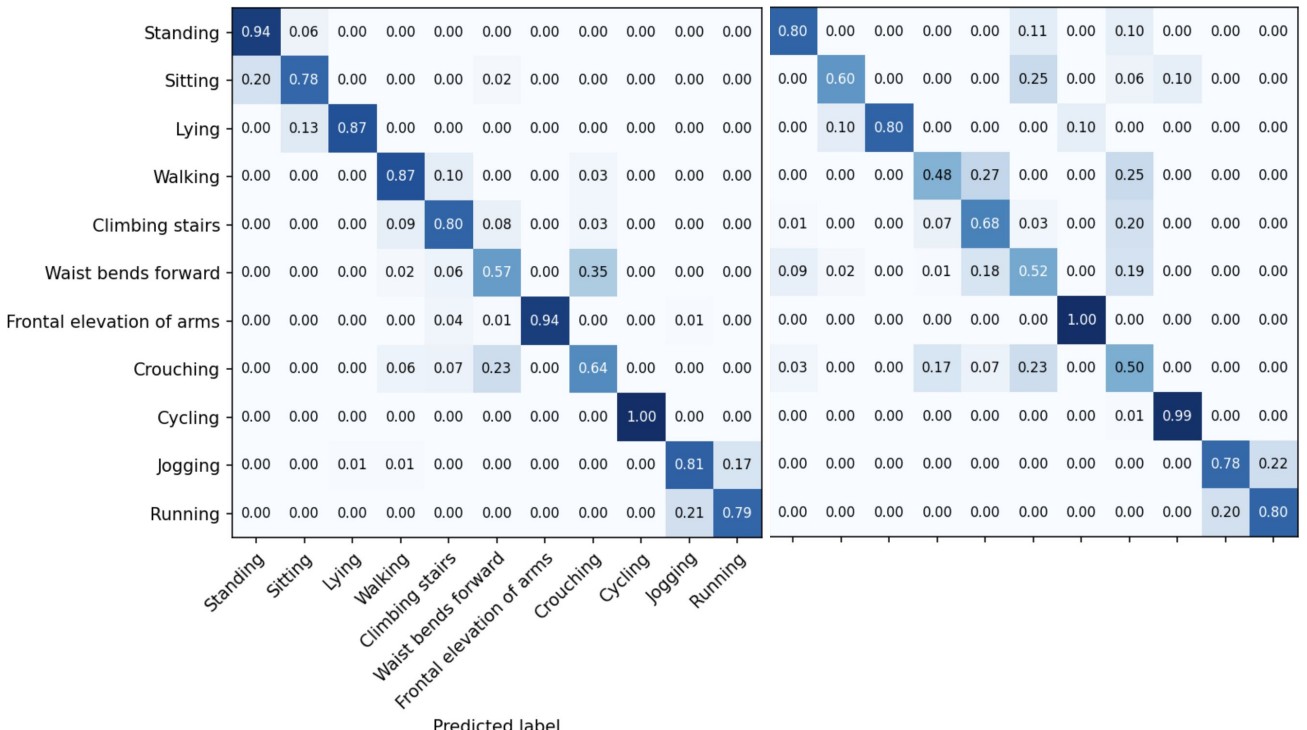

*Figure 9.* Confusion matrices for Bio-PM (left) vs TF-C (right) on the Mhealth dataset.

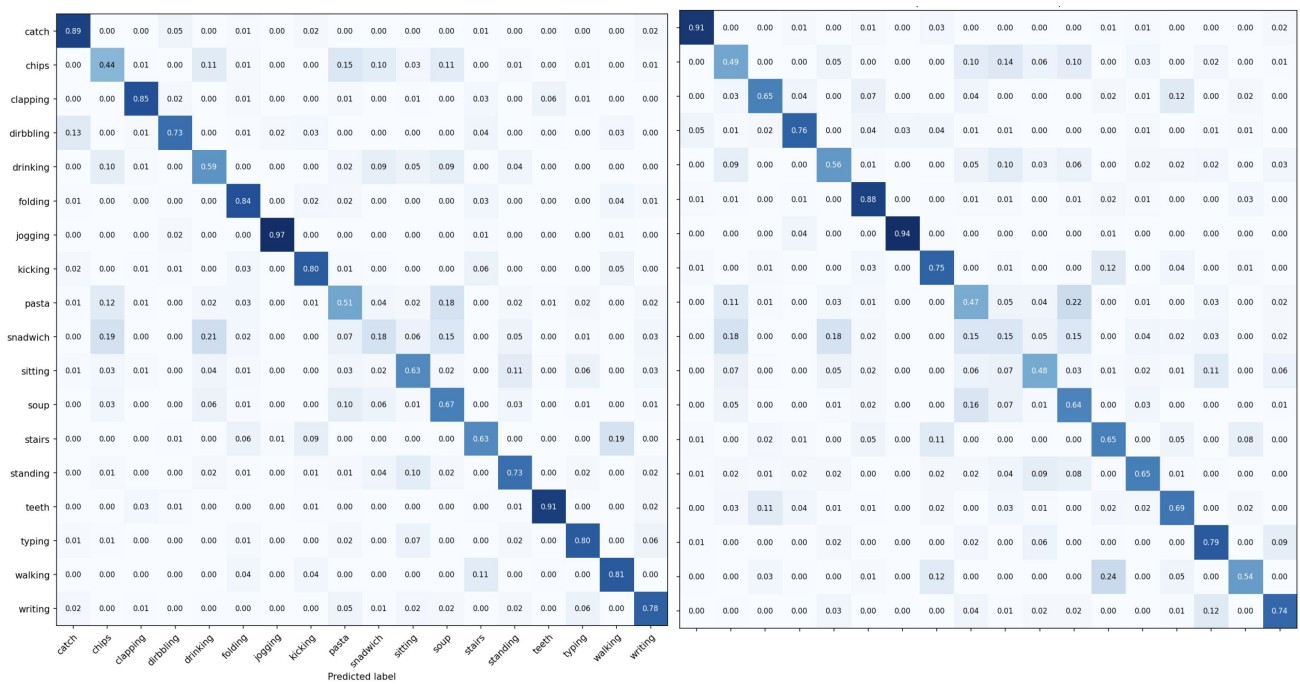

*Figure 10.* Confusion matrices for Bio-PM (left) vs TF-C (right) on the WISDM dataset.

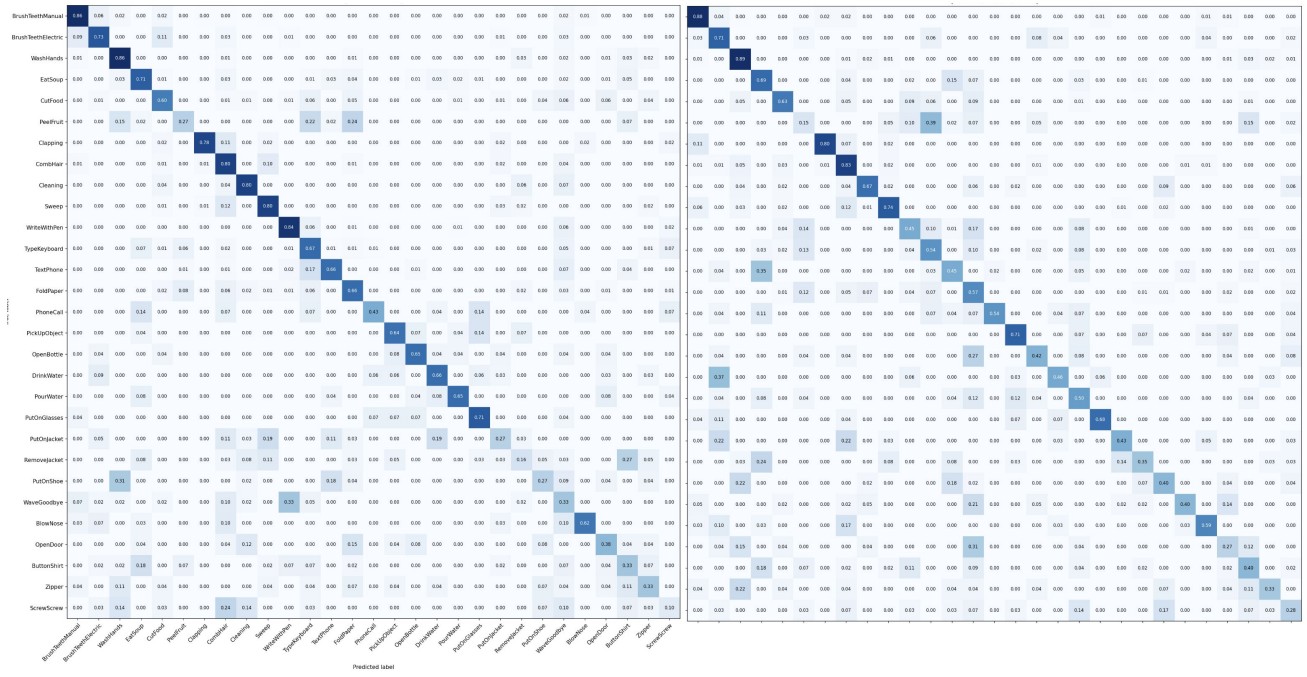

*Figure 11.* Confusion matrices for Bio-PM (left) vs TF-C (right) on the UMAHand dataset.

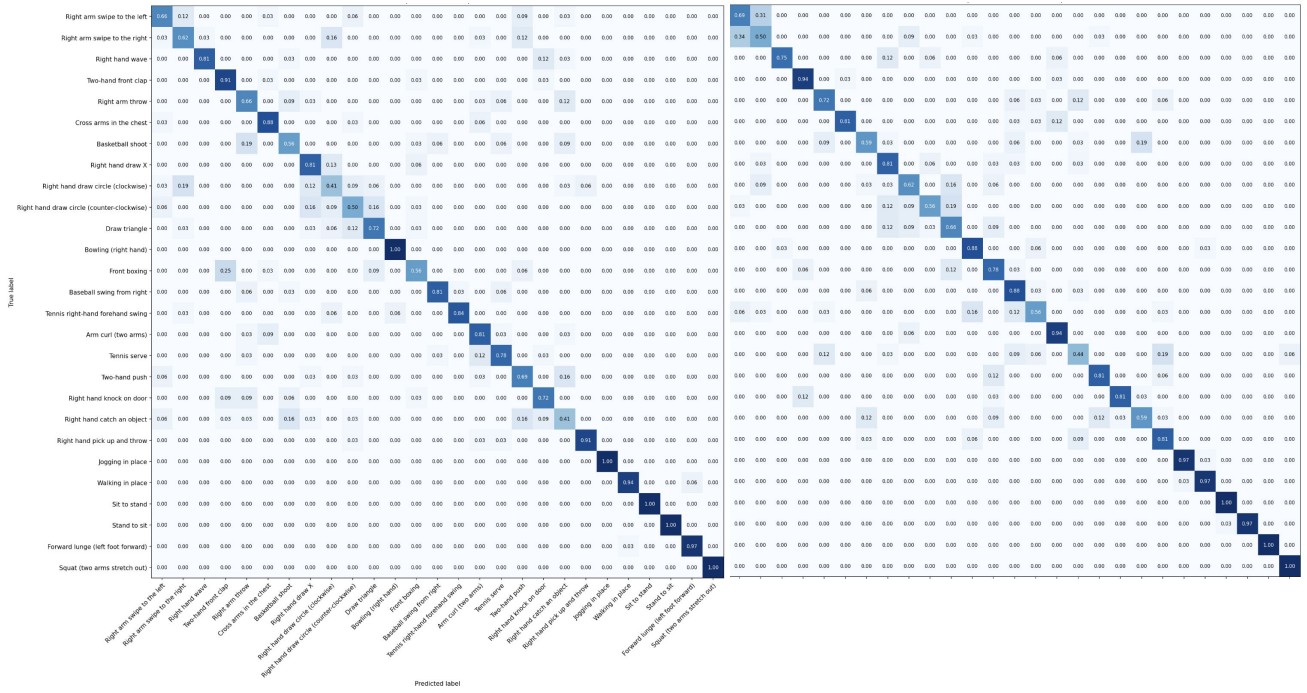

*Figure 12.* Confusion matrices for Bio-PM (left) vs TF-C (right) on the HAD dataset.

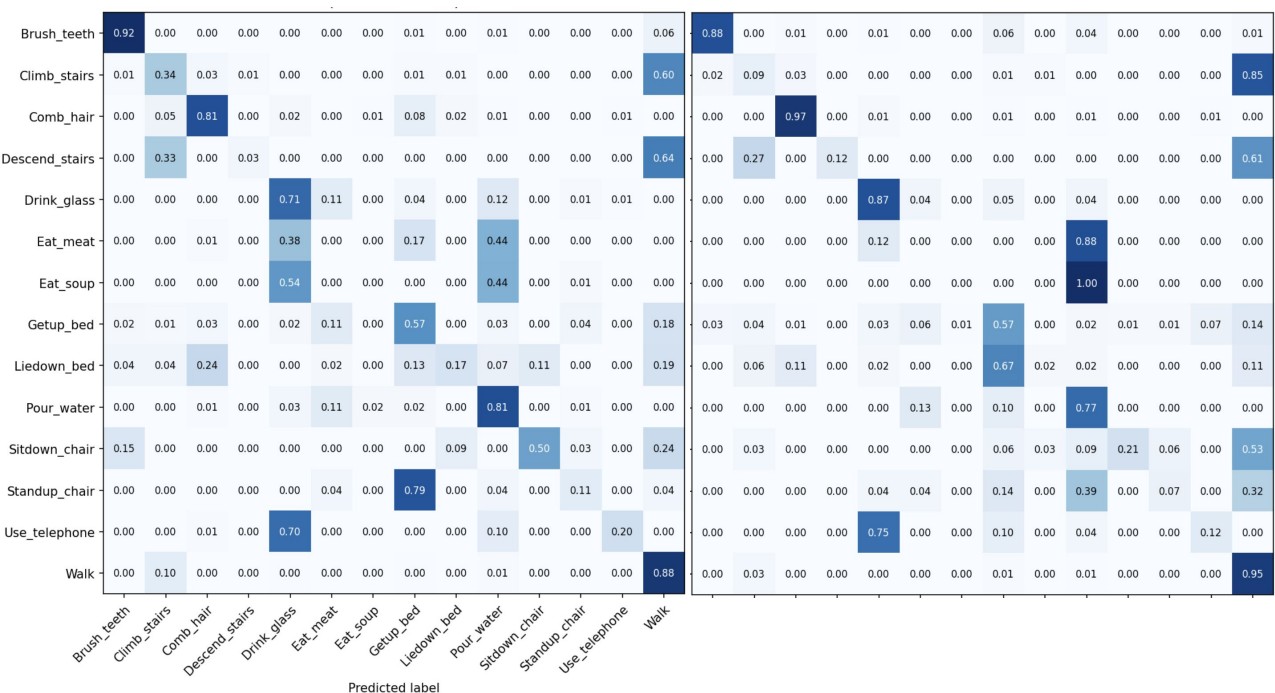

*Figure 13.* Confusion matrices for Bio-PM (left) vs TF-C (right) on the WHARF dataset.

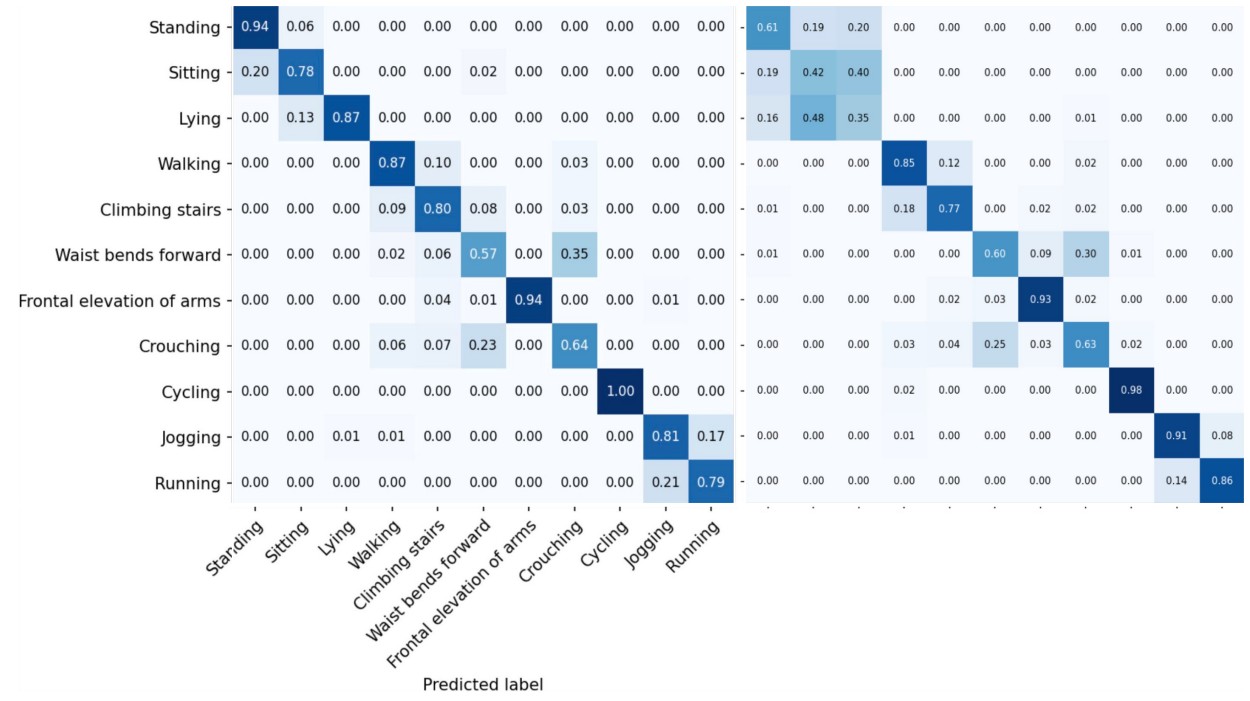

*Figure 14.* Confusion matrices for Bio-PM (left) vs Bio-PM without gravity (right) on the Mhealth dataset.

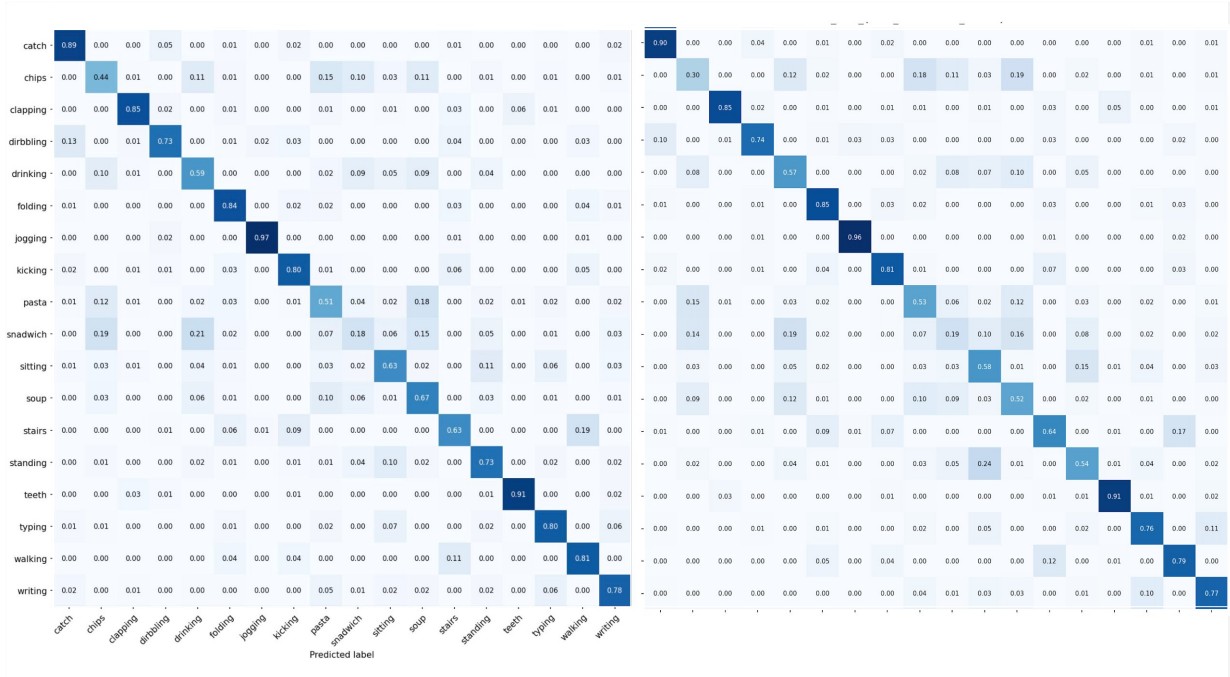

*Figure 15.* Confusion matrices for Bio-PM (left) vs Bio-PM without gravity (right) on the WISDM dataset.

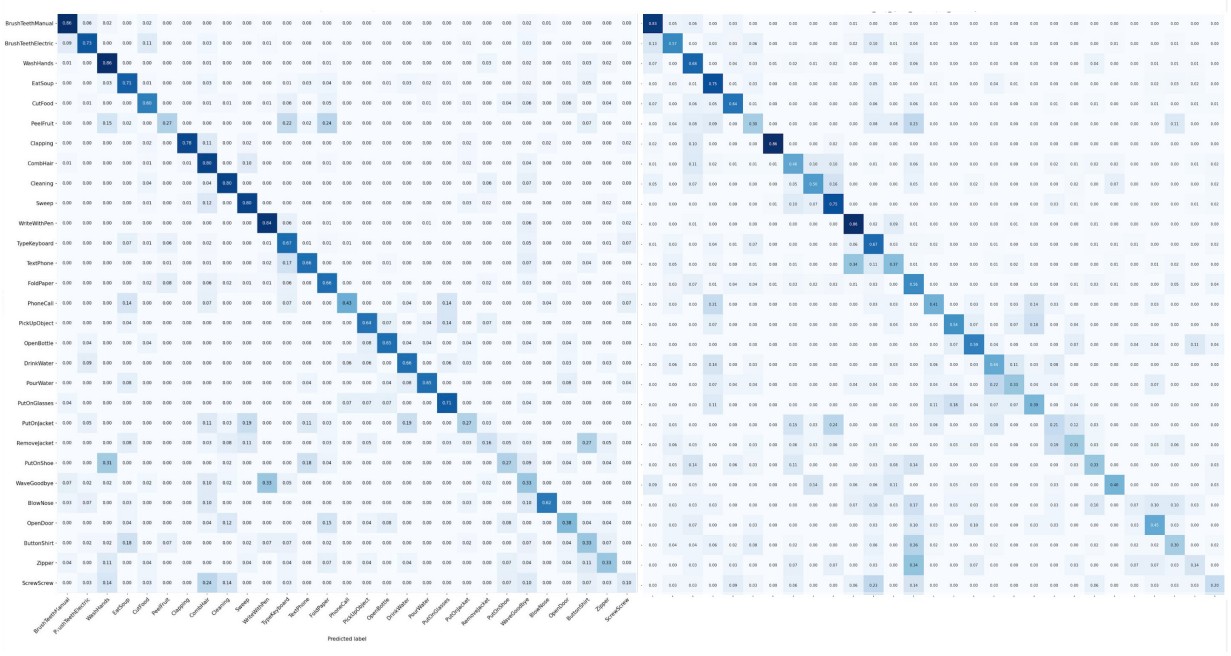

*Figure 16.* Confusion matrices for Bio-PM (left) vs Bio-PM without gravity (right) on the UMAHand dataset.

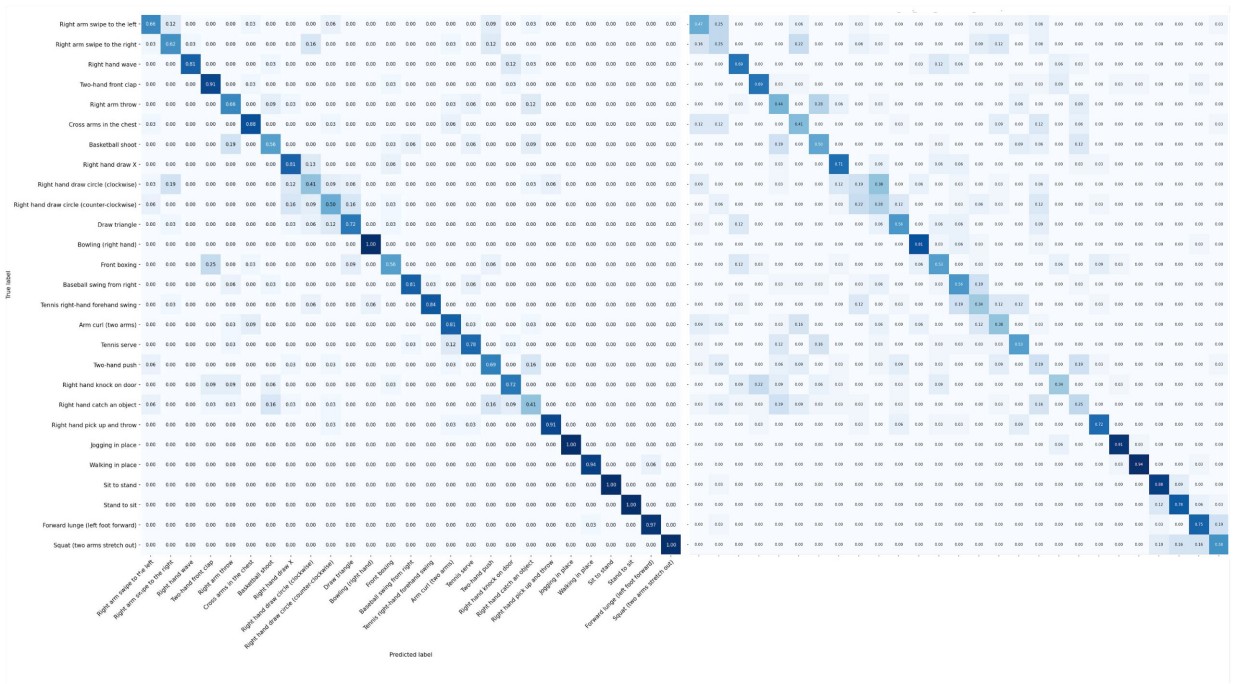

*Figure 17.* Confusion matrices for Bio-PM (left) vs Bio-PM without gravity (right) on the HAD dataset.

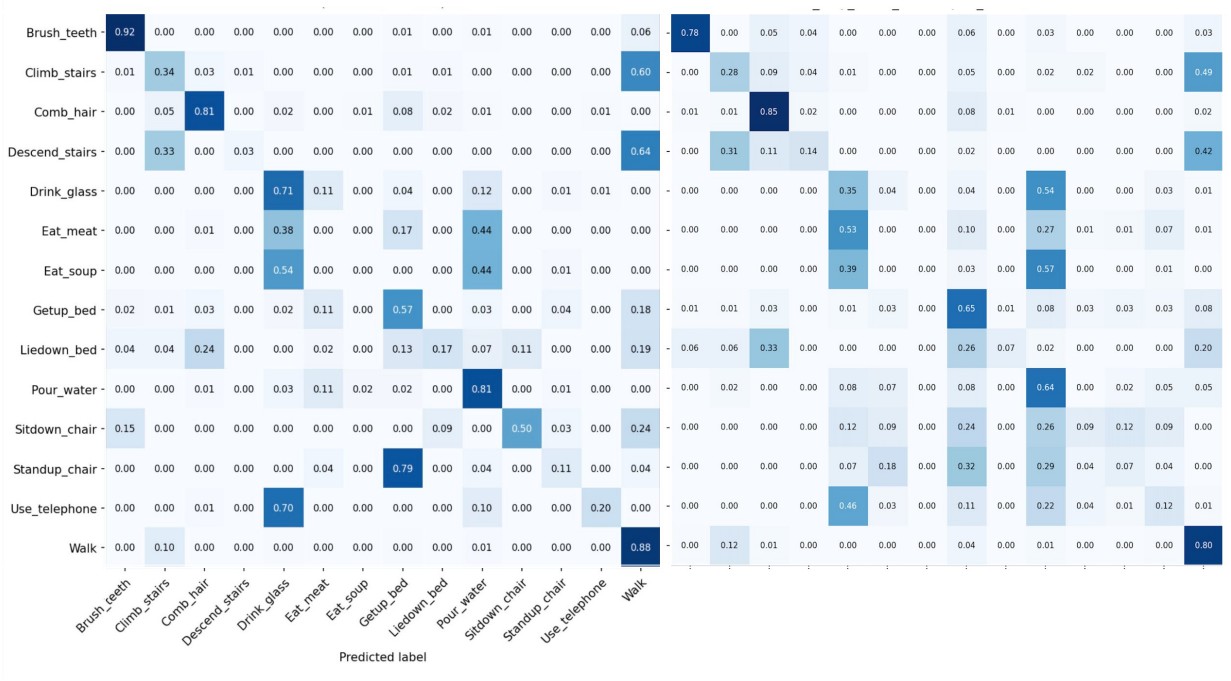

*Figure 18.* Confusion matrices for Bio-PM (left) vs Bio-PM without gravity (right) on the WHARF dataset.

