# OpenReview forum: "Bio-Inspired Self-Supervised Learning for Wrist-worn Accelerometer Data"
_ICML.cc/2026/Conference — ICML 2026 regular_

### Official Review · Reviewer_iYT6 · 2026-03-04

**Soundness:** 2
**Presentation:** 2
**Significance:** 1
**Originality:** 4
**Overall Recommendation:** 4
**Confidence:** 5

**Summary:**

This paper introduces Bio-PM, a self-supervised learning (SSL) approach for Human Activity Recognition (HAR) using wrist-worn accelerometer data. The core contribution is a bio-inspired tokenization strategy grounded in the "submovement theory" of motor control. Instead of chunking time-series data into arbitrary fixed-length windows, the authors define tokens based on "movement segments" bounded by zero-crossings in the gravity-reduced acceleration signal. Using these biologically motivated tokens, the authors pretrain a Transformer encoder via masked segment reconstruction on the large-scale NHANES dataset. The learned representations are evaluated via linear probing on six subject-disjoint HAR benchmarks, demonstrating superior performance and data efficiency compared to strong wearable SSL baselines (e.g., TF-C, AugPred) trained under identical conditions.

**Compliance With Llm Reviewing Policy:**

Affirmed.

**Ethical Review Concerns:**

None.

**Final Justification:**

Please refer to the Rebuttal Acknowledgement.

**Key Questions For Authors:**

### 1. Regarding the task itself.
The main task seems to be HAR using data from a single wrist-worn IMU (or smartwatch). I think there are three areas here that need better explanation.
First, the specific motivation for this task isn't explained clearly enough, and the related work section feels a bit incomplete. This makes the overall narrative feel somewhat disconnected. Second, the title explicitly says "IMU Signals", but the method only uses accelerometer data. This discrepancy needs an explanation. Third—and this is a key question for me because of the missing context in my first point—how important is this task itself? I'm familiar with IMU and inertial-based motion recognition/estimation, where we typically leverage the lightweight and lighting-robust nature of IMUs to replace optical/vision-based methods in specific scenarios. So, why exactly is wrist-only IMU useful for HAR? The authors mention applications like medical analysis, but I didn't find any concrete examples in the paper, nor did the authors provide any demo videos. Furthermore, I'm pretty skeptical about the effectiveness of using just a single IMU (in this case, a single accelerometer). IMU signals are inherently sparse, and accelerometer data suffers from drift and noise. On top of that, wrist acceleration happens in a non-inertial frame. For example, if a person spins in place while keeping their wrist still relative to their torso, it generates centripetal acceleration. If the person's torso is perfectly still but they just rotate their wrist, it can generate the same acceleration profile. With your setup, there is no way to distinguish between the two. Also, the accelerometer signals are inheretly drifty and noisy. I would actually argue that these physical interference issues are the real limiting factors for wrist-based HAR, rather than just a lack of labeled data. This part is my biggest concern and I'd love to hear how the authors view these issues.


### 2. Regarding the theoretical foundation.
The theoretical foundation sounds really interesting, but right now it feels like it's just being thrown in as an abstract assumption. I was hoping to see a deeper, more intuitive explanation. For instance, why exactly does this theory translate well to HAR or SSL? Could you provide some concrete examples showing how tokenizing the acceleration signal actually improves the representation quality? More specifically, how and why can the "submovement theory of motor control" be directly interpreted as the token segmentation of acceleration signals in this specific context?

### 3. Regarding the claim.
"largely due to the imperfect separation of digital filtering, it captures not only static gravity but also slow rotational movements, providing essential postural context (e.g., distinguishing sitting from lying)." This is a major claim, and it requires further justification, both theoretically and experimentally.

**Limitations:**

The authors briefly discuss scaling to larger datasets and clinical endpoints in Section 5. However, they have not adequately discussed the fundamental limitations and potential failure modes of relying exclusively on a single wrist-worn accelerometer, particularly regarding orientation drift and centripetal ambiguities. I suggest expanding the limitations section to address the boundaries of accelerometer-only inference.

**Strengths And Weaknesses:**

### Strengths

The application of the submovement motor control theory to address the tokenization bottleneck in continuous time-series SSL is creative. Bridging low-level kinematics with high-level representation learning offers a fresh perspective.

The empirical evaluation is rigorously controlled. By pretraining all baseline methods on the exact same NHANES corpus and applying identical downstream linear-probing protocols, the authors successfully isolate the impact of their tokenization and sequence modeling strategies.

### Weaknesses

There is a discrepancy between the title and the methodology. The title claims to learn from "IMU Signals", but the method explicitly uses only a "wrist-worn triaxial accelerometer". An IMU inherently implies the inclusion of a gyroscope (and a magnetometer for 9-axis IMU). Accelerometer-only sensing suffers from severe physics-based ambiguities in non-inertial frames (e.g., distinguishing centripetal acceleration during wrist rotation from actual linear displacement without gyroscope data). The paper fails to acknowledge or address how this fundamental sensing limitation affects the proposed HAR task. I will elaberate on this in the "key questions" section.

The authors claim that the low-frequency component captures "essential postural context (e.g., distinguishing sitting from lying)". While Table 2 empirically shows an overall performance drop without gravity, the claim lacks justification or targeted ablation to prove which specific postural transitions are being successfully resolved by this low-pass filter, especially given the aforementioned single-sensor ambiguities.

The visual presentation falls short of conference standards. The font sizes in Figures 1, 2, and 3 are inconsistent, often too small to read comfortably, and do not utilize the available whitespace efficiently. Conversely, the fonts in some appendix figures appear disproportionately large. The related work and introduction also lack concrete, illustrative examples (or demos) of why wrist-only motion recognition remains a critical challenge in specific medical or wellness contexts beyond broad claims.

---

> ### Author Rebuttal · Authors · 2026-03-30
>
> We thank the reviewer for recognizing the creativity of our bio-inspired tokenization and the rigor of our evaluation.
>
> **Title and figures.** We agree the title should be more precise and will revise it to include ‘Accelerometer Signals’, and standardize font sizes across Figures 1–3.
>
> **Why wrist-only accelerometry?** We do not claim a single wrist accelerometer resolves all inertial ambiguities. While multi-sensor inertial systems offer superior kinematic fidelity, single-wrist sensing—specifically accelerometry—is the most viable modality for long-term health monitoring due to low social and user burden. While gyroscopes add critical orientation information, they draw up to 10x more power (Sucerquia et al., 2018, PMID: 29621156). Consequently, the field’s most significant unlabeled corpora (UK Biobank, NHANES) and labeled benchmarks (WISDM, PAMAP2) consist exclusively of wrist accelerometer data. NHANES explicitly excluded gyroscopes to prioritize extended battery life ([CDC link](https://wwwn.cdc.gov/Nchs/Data/Nhanes/Public/2011/DataFiles/PAX80_G.htm)).
>
> Significant research has focused on SSL for wrist accelerometer-based HAR because it is the fundamental task for digital health monitoring (Xu et al., 2025, arXiv:2411.18822v5; Yuan et al., 2024, DOI:10.1038/s41746-024-01062-3; Haresamudram et al., 2022, arXiv:2202.12938). Thus, we aim to advance representation learning *within this framework*, demonstrating superior results compared to existing SSL methods.
>
> Crucially, despite inherent limitations, wrist-worn accelerometry has shown clinical efficacy across diverse domains, such as assessing disease progression in Amyotrophic Lateral Sclerosis (Gupta et al., 2023, PMID: 37604821) and identifying prodromal symptoms in Parkinson’s (Schalkamp et al., 2023, PMID: 37400639).
>
> We will extend the Related Works to clarify this, and the Limitations section to explicitly address single-sensor physical ambiguities.
>
> ---
>
> **Theoretical foundation.** Research has shown tokenization represents a foundational inductive bias in sequence modeling (Rajaraman et al., 2024, DOI: 10.52202/079017-1999; Gastaldi et al., 2025, arXiv:2407.11606). Specifically, tokenizations that better respect the underlying structure of the domain substantially improve performance (Schmidt et al., 2024, arXiv:2402.18376; Spathis & Kawsar et al., 2024, PMID: 38950417), allowing models to prioritize relationships **between coherent events** rather than wasting capacity recovering this structure from arbitrary fragments.
>
> Building on this paradigm, our work introduces a new token for wrist accelerometry called movement segments, motivated by the submovement theory. We hypothesize that by learning the compositional organization of these meaningful motor units from large-scale, unlabeled data, we can derive latent representations that more effectively distinguish human activities. Thus, our approach is a hypothesis-driven, empirically validated framework.
>
> We demonstrate our method yields superior representation quality via linear probing compared to SOTA SSL methods. While a closed-form interpretation of Transformer embeddings remains an ongoing community challenge, our results show that this tokenization specifically improves discrimination between fine-grained, similar activity groups (stairs vs. walking; tennis serve vs. right-hand throw; or various eating/drinking gestures in MHealth/WISDM/WHARF). By shifting the learning objective from raw time-series morphology to the organizational structure of movement segments, we believe our model overcomes the traditional difficulties of differentiating activities with overlapping signal characteristics.
>
> We will add this to the revision.
>
> ---
>
> **Regarding the gravity claim.** Our intention was to specify that the low-frequency vector within the sensor’s local coordinate system provides essential postural data during “sedentary activities.” This is based on the principle that gravity remains quasi-static within accelerometry, as wrist rotations typically occur below 0.5 Hz (van Hees et al., 2013, PMID: 23626718). Thus, this vector provides the essential information needed to distinguish between sedentary postures (e.g., sitting vs. lying) that may otherwise appear identical in high-frequency linear acceleration. We will clarify these points in the revised manuscript.
>
> Moreover, we will add new empirical evidence, comparing per-class confusion matrices (Figures [*at this link*](https://drive.google.com/open?id=1X5bcm2sqMyWB5qzUIO52NwLj2x1pzSXS)) for Bio-PM with vs. without the gravity component. Removing gravity degraded postural discrimination while leaving dynamic activities intact. The clearest case is mHealth, where Standing drops from 0.94 to 0.61, Sitting from 0.78 to 0.42, and Lying from 0.87 to 0.35, with the errors concentrating among these three classes. Similar patterns were observed in WISDM and WHARF.
>
> ---
>
> We hope these clarifications and new analyses fully address your concerns.

---

> > ### Author Rebuttal · Reviewer_iYT6 · 2026-04-02
> >
> > Thank you to the authors for the detailed rebuttal. My feedback is as follows:
> >
> > **1. Regarding the task scope and related work:**
> > I appreciate the authors for supplementing the related work on wrist accelerometer health analysis and committing to revising the title and figures. I have no further concerns here.
> >
> > **2. Theoretical foundation and tokenization mechanism:**
> > I must admit that the logic in the rebuttal regarding this point was somewhat disconnected from my original question. Reiterating that tokenization is a proven inductive bias in NLP or that the framework is "hypothesis-driven and empirically validated" does not mechanistically answer *why* exactly this theory translates well to wearable HAR or SSL.
> >
> > However, upon revisiting Section 3.2 of the manuscript, I found that the manuscript states that while submovements are inherently bell-shaped velocity profiles, directly adapting this in the velocity domain poses practical hurdles (e.g., integration drift). Thus, the kinematic translation to define token boundaries at acceleration zero-crossings bridges the biological theory and practical signal processing.
> >
> > Furthermore, the ablation study (Table 2: Bio-PM w/o Proposed Tokenization vs. Naive Tokenization) supports that this zero-crossing alignment is perhaps the most critical component of the proposed method, empirically validating the mechanism behind the theory. I have no further concerns on this part, but I think emphasizing this part could benifit the paper, as it constitutes one of the most significant technical contributions of this work.
> >
> > **3. The gravity claim:**
> > Thank you for providing the additional results. They offer evidence that incorporating the low-frequency (gravity) component resolves certain pose ambiguity. However, while gravity coordinates solve ambiguities between orthogonal postures, they might still struggle to differentiate "Sitting" and "Standing" if the user's arm is hanging down vertically. Nevertheless, I think this is an inherent limitation of single-wrist sensing and might fall beyond the scope of the current submission.
> >
> > Overall, in light of the clarifications and the planned revisions, I believe the strengths of this paper outweigh its initial weaknesses. I am happy to raise my score to a Weak Accept.

---

### Official Review · Reviewer_2QTy · 2026-03-11

**Soundness:** 3
**Presentation:** 3
**Significance:** 3
**Originality:** 3
**Overall Recommendation:** 4
**Confidence:** 3

**Summary:**

In this paper, the authors proposed to use the submovement theory to guide the training of tokenized accelerometry. The author pre-trained the IMU data with a Transformer under masked reconstruction. The author then validates the effectiveness of the pre-trained IMU model on downstream HAR tasks using linear probing.

**Compliance With Llm Reviewing Policy:**

Affirmed.

**Key Questions For Authors:**

1. Why did the author choose linear probing, but not other approaches such as fine-tuning or PEFT? Would the results be fair to compare with only linear probing?
2. How noise-robust is the approach? Since the zero-crossing rate has been long studied in fields of speech processing, and apparently the noise will impact the zero-crossing rule.
3. I am not sure about the term Temporal reasoning (Transformer) used in the paper. Can the author explain why or how it is reasoning?

**Limitations:**

1. I think the authors should also note the limitations of the use of the standardized HAR dataset in downstream prediction. These datasets have repetitive and structured activities, and may not be generalized to real-world behaviors.

**Strengths And Weaknesses:**

Strength of the paper:
1. The paper is well-written and easy to understand, with nice figures presenting the approach
2. The paper introduces a bio-inspired approach in existing human kinetics for modeling IMU data, which is very interesting
3. The paper pre-trains the IMU data with the proposed approach on a large-scale IMU dataset

Weakness of the paper:
1. Using linear probing alone might not be a fair comparison
2. It is unclear how noise-robust the approach is. For example, the zero-crossing rate has long been studied in fields of speech processing, and apparently, the noise will impact the zero-crossing rule.

---

> ### Author Rebuttal · Authors · 2026-03-30
>
> We thank the reviewer for recognizing that our work is well presented, our bio-inspired tokenization is interesting, and for acknowledging the scale of our pretraining data.
>
> **Q1: Why Linear Probing?** We thank the reviewer for this question. The central question of this work is whether our tokenization produces richer latent representations compared to existing SSL approaches based on fixed-window tokenization. We selected linear probing as our evaluation metric because it directly measures what SSL pretraining *already* encodes without any task-specific adaptation.
> Using linear probing ensures architectural fairness. By freezing the backbone, we evaluate the quality of the pretrained representations themselves rather than the model's varying capacity to compensate for weaker features during supervised fine-tuning. Because the baseline architectures range from 1.4M to 10.5M parameters, methods like fine-tuning or PEFT may conflate representation quality with their adaptation capacities. To further fairly isolate the effect of tokenization, we pretrained all SSL baselines on the same large-scale NHANES corpus rather than using off-the-shelf checkpoints.
> Finally, we note that this evaluation strategy is validated empirically. A frozen Bio-PM with a linear head (~8k–26k trainable parameters depending on dataset) outperforms the same architecture trained from scratch with all 1.4M parameters trainable (0.65 vs. 0.55 Macro-F1; Table 2). This result indicates that Bio-PM’s pretrained representations capture transferable structure that the same architecture cannot recover from limited labeled data when trained from scratch, which is precisely the property linear probing is intended to test.
>
> **Q2: How noise-robust is the approach?** We sincerely thank the reviewer for this useful suggestion; addressing it has directly strengthened our manuscript.
>
> High noise in the accelerometer signal could indeed lead to spurious zero-crossings. Therefore, as detailed in Section 3.3 and Appendix F.2, our tokenization employs a two-stage hysteresis filter to mitigate spurious tokens. Specifically, zero-crossing segments with a duration shorter than 50 ms or an amplitude below 0.01 g are not recognized as independent movements; instead, they are merged into the subsequent segment that meets these thresholds. This filtering ensures that insignificant signal fluctuations caused by noise are ignored, allowing the model to focus on the systematic organization of prominent movement patterns.
> To empirically demonstrate the robustness of our method, we conducted an additional experiment by injecting AWGN into the input accelerometer signals. This choice is grounded in prior research, which confirms that AWGN effectively models the noise characteristics typically found in accelerometry (Nirmal et al., 2016, DOI: 10.1117/12.2234255). We simulated various noise levels—approximately 27 dB, 21 dB, 15 dB, 9 dB, and 3 dB—which translate to noise floors roughly 1.5x, 2x, 3x, 5x, and 9x higher than typical levels.
>
> As shown in the Figure [*at this link*](https://drive.google.com/open?id=1WHS6h2204J69-lYgwG5RDh65czazU_du), performance remains broadly robust until approximately 15 dB (3x typical noise) and only degrades significantly thereafter in the range typically characterized by poor SNR. Across HAD, mHealth, and PAMAP2, the Macro-F1 score is largely preserved through the 21–15 dB range, with only modest degradation at 9 dB and a clearer drop-off only at the extreme 3 dB setting. These results indicate that Bio-PM is robust to realistic increases in sensor noise, and that the combination of hysteresis-based tokenization and large-scale pretraining yields representations that remain informative even under substantial noise amplification.
>
> We will include these results in a new subsection `4.5 Robustness`.
>
> **Q3: Temporal Reasoning.** Thank you for catching that. We intended to refer to the Transformer's ability to model temporal dependencies across movement segments. We will rename this heading to "Temporal Modeling" in the revision.
>
> ---
>
> **Limitations:** We agree and will add this as an explicit limitation. While our benchmarks cover substantial variety — UMAHand includes 29 fine-grained hand activities and WISDM includes 18 diverse daily activities (see confusion matrices in Appendix H) — we acknowledge that generalization to fully unstructured free-living behavior remains untested.
>
> ---
>
> We thank the reviewer for their thoughtful questions and feedback; it prompted a valuable new empirical analysis that directly strengthened our manuscript.

---

> > ### Author Rebuttal · Reviewer_2QTy · 2026-04-02
> >
> > The author has answered most questions, and I will keep my score.

---

### Official Review · Reviewer_TiUk · 2026-03-11

**Soundness:** 3
**Presentation:** 3
**Significance:** 3
**Originality:** 3
**Overall Recommendation:** 5
**Confidence:** 4

**Summary:**

The paper proposes a tokenization strategy for wrist-worn accelerometer signals grounded in the submovement theory of motor control. Instead of chopping sensor streams into fixed-length windows, the authors define tokens as the intervals between consecutive acceleration zero-crossings — movement segments that they argue act as word-like units of human motion. These segments are encoded by a CNN, augmented with metadata, and fed into a Transformer pretrained via masked reconstruction on a large public corpus. The pretrained encoder, Bio-PM, is evaluated across multiple subject-disjoint HAR benchmarks with frozen linear probing, consistently outperforming controlled SSL baselines. Additional experiments show improved label efficiency and generalization to unseen token transitions.

**Compliance With Llm Reviewing Policy:**

Affirmed.

**Key Questions For Authors:**

1. Have you run paired tests across folds for the comparison with the strongest baseline? If per-dataset significance is weak but the aggregate trend is reliable, stating this explicitly would be more transparent.
2. How sensitive is the tokenization to the high-pass filter cutoff used to separate voluntary motion from gravity? Slow movements or postural transitions might be affected, and even a small sweep would be reassuring.

**Limitations:**

The authors openly discuss the need for larger upstream corpora, extension to clinical endpoints, and generalization to other modalities. The impact statement is appropriate.

**Strengths And Weaknesses:**

Strengths

Borrowing the submovement framework from motor control to design tokens for wearable SSL is a fresh angle that I haven't encountered in prior HAR work. The analogy to words in language modeling gives the reader a concrete mental model.

The controlled protocol deserves credit — all baselines share the same upstream corpus, windowing, and evaluation pipeline, so downstream differences are easier to attribute to the objective itself. This level of control is rare in wearable SSL papers.

The tokenization ablation is the most convincing piece of evidence. Swapping only the segmentation strategy while holding everything else fixed produces a clear drop, making a strong case that movement-aligned boundaries provide a useful inductive bias.

The unseen-transition probe adds a nice diagnostic layer. The shuffle control confirms the model relies on sequential ordering rather than token identity alone.

Writing is clear, the pipeline is easy to follow, and the appendix is detailed.

Weaknesses

On several benchmarks the improvement over the strongest controlled baseline is modest, with overlapping standard deviations. No paired statistical test is reported, so it is hard to tell whether per-dataset gains are reliably above noise. Even a brief discussion of this would help.

Evaluation is limited to frozen linear probing. A "no-pretraining" ablation is included but it fine-tunes from scratch, which is a different question from "pretrain then fine-tune." Fine-tuning numbers for all methods would round out the picture.

The approach is specific to wrist accelerometry and upper-limb kinematics. The authors note this, but it does constrain broader impact.

Some appendix ablations cover only a subset of benchmarks without stating why. The computational cost of the zero-crossing pipeline versus naive chunking is also not mentioned.

---

> ### Author Rebuttal · Authors · 2026-03-30
>
> We thank the reviewer for recognizing that borrowing from submovement theory is “a fresh angle,” and especially that the level of control in our evaluation is rare in wearable SSL.
>
> **Q1. Regarding the statistical significance of our results:** We performed one-sided Wilcoxon signed-rank tests (Bio-PM > TF-C) both per dataset and pooled across all 41 folds (see Table [*at this link*](https://drive.google.com/uc?export=view&id=1ui4N6amh_YgKq0W-NrR6kl_-FJ_5o9MG)).
>
> Individually, results for mHealth and WISDM reached statistical significance (p < 0.05), while UMH (p = 0.063) and PAMAP (p = 0.074) approached significance. WHARF and HAD did not reach significance (p > 0.05). This variance is expected given the limited statistical power inherent in the smaller fold counts of these specific benchmarks.
> The most compelling evidence lies in the aggregate trend, which provides a sufficient sample size for a robust statistical test. Bio-PM outperforms TF-C in 31 of 41 total folds, yielding a pooled Wilcoxon result of p < 0.0001 with a Cohen’s d of 0.61 (a medium effect size). We agree that explicitly stating this aggregate reliability provides a more transparent and robust empirical narrative, and we will include the table and analysis in the revised manuscript.
>
>
> **Q2. Regarding sweeping high-pass filter cutoff:** This is an excellent question. While sweeping the cutoff for Bio-PM's pretraining would require re-pretraining from scratch for each value—which is computationally prohibitive—we directly tested what matters most for the reviewer's concern: whether downstream performance is fragile to the gravity-separation threshold. We swept the high-pass cutoff at downstream probe time (0.2–1.0 Hz) while keeping the pretrained encoder fixed (pretrained at 0.5 Hz), see Figure [*at this link*](https://drive.google.com/open?id=1ix_lmxZeZJL3DF2aW6Lpaq9ykqvZPyZw).
>
> Across four benchmarks, Macro-F1 varies by only a few points, remaining well within standard deviations with no consistent directional trend. This confirms that the learned representations are robust to the precise gravity-separation threshold. We will add this ablation to a new subsection `4.5 Robustness`.
>
> ---
>
> **Additional Clarifications:**
>
> **1. Linear Probing:** We chose frozen linear probing because it isolates the quality of the representations pretraining *already* encodes, and does not confound the comparison with differences from fine-tuning dynamics across architectures of varying capacity (1.4M–10.5M parameters). For a more detailed discussion, we refer the reviewer to our response to Reviewer 3 (2QTy). We will acknowledge that end-to-end fine-tuning remains a natural extension of this work in the revised manuscript.
>
> **2: Is the approach specific to wrist accelerometry?** We appreciate the reviewer for raising this point and allowing us to clarify the broader applicability of our work. While this study validates the proposed tokenization specifically on wrist-worn accelerometer data, the approach is inherently extensible to other inertial modalities and body locations. The submovement theory introduced by Hogan and Sternad applies not only to linear kinematics but also more broadly to the angular kinematics of joints (Hogan & Sternad, 2012); thus, it theoretically extends to gyroscope and magnetometer data, where angular and rotational velocity can be described as a superposition of bell-shaped primitives. Furthermore, while we focused on wrist-worn sensing—the most common and preferred configuration for long-term monitoring—prior evidence indicates that submovement theory applies to other end-effectors, including the chest, head, and ankles (Michmizos et al., 2014; PMID: 25505881, Chen et al., 2012; PMID: 23139749, de Lemos Fonseca et al., 2020; PMID: 31846518, Silva et al., 2026; j.bspc.2026.110099). We will include this discussion in the revised manuscript to highlight the broader impact and potential future directions of our work.
>
> **3. Why do some ablations cover only a subset of benchmarks?** We appreciate the reviewer’s thorough assessment of our Appendix. This decision was mainly because (1) of computational cost - WISDM (51 subjects) is substantially more expensive to run across multiple ablations, and (2) UMH was added after initial experimentation, when we already had consistent signals from the remaining datasets. We will clarify this in the revision.
>
> **4. Computational cost of the zero-crossing pipeline.** We appreciate the reviewer for flagging this point. On a single CPU core, tokenizing a batch of 32 MHealth windows takes ~0.4 s with our pipeline vs. ~0.15 s for naive chunking, a ~2.7× overhead that is modest given the downstream gains. We will add this comparison to the revised paper.
>
> ---
>
> We sincerely thank the reviewer for their careful reading. Your specific questions prompted valuable new statistical and robustness analyses that directly strengthened our manuscript.

---

> > ### Author Rebuttal · Reviewer_TiUk · 2026-04-07
> >
> > My concerns have been addressed, I will keep my positive score.

---

### Official Review · Reviewer_VL8G · 2026-03-12

**Soundness:** 4
**Presentation:** 3
**Significance:** 4
**Originality:** 4
**Overall Recommendation:** 5
**Confidence:** 3

**Summary:**

This paper proposes Bio-PM, a self-supervised pretraining framework for human activity recognition (HAR) from wrist-worn IMU data, grounded in submovement theory from motor control. The authors argue that tokenization is a key bottleneck in wearable SSL: conventional fixed-length window segmentation does not align well with the structure of human motion. Instead, Bio-PM segments IMU signals into biologically inspired “submovements,” defined using acceleration zero-crossings to approximate point-to-point wrist motions. These movement segments are then treated as tokens for self-supervised learning.

Pretraining is performed via masked segment modeling on a large unlabeled IMU corpus, where entire movement segments are masked and reconstructed. The learned representations are evaluated through downstream transfer across six HAR datasets and compared against five alternative SSL objectives. Bio-PM consistently achieves superior transfer performance, suggesting that motion-aligned tokenization improves representation quality in wearable sensor SSL settings.

**Compliance With Llm Reviewing Policy:**

Affirmed.

**Final Justification:**

The authors have thoroughly addressed the concerns raised in the review. The clarification of generalization beyond point-to-point motions, the discussion of applicability to other sensing modalities, and the per-class performance analysis strengthen the empirical interpretation of the results. The acknowledgement of dataset bias and privacy considerations also addresses the limitations raised. I maintain my original assessment.

**Key Questions For Authors:**

1.	Bio-PM is grounded in submovement theory, which assumes point-to-point wrist motions. To what extent do the six benchmark datasets reflect motion patterns that conform to this assumption? Have the authors analyzed whether segmentation quality or performance gains vary across datasets with differing activity types?

**Limitations:**

As a grammar of movement, it's reasonable to wonder if there may be cultural aspects that are biased for or against, given the training sets. There may be a possibility that tokenization could be re-indentify in certain ways.

**Strengths And Weaknesses:**

**Strengths**

The paper presents a clearly motivated and conceptually original approach to self-supervised learning for wrist-worn IMU signals. By grounding tokenization in submovement theory from motor control, the authors introduce a biologically informed segmentation strategy that reframes motion primitives as meaningful tokens for SSL. This cross-domain conceptual bridge between motor control theory and representation learning is novel and well articulated.

The empirical evaluation is thorough within the HAR setting, including comparisons against five alternative SSL objectives and transfer experiments across six datasets. The proposed method consistently outperforms competing pretraining strategies, suggesting that movement-aligned tokenization improves representation quality. The theoretical motivation aligns well with the experimental results, reinforcing the plausibility of the “grammar of movement” hypothesis as a structural prior for wearable sensor learning.

**Weaknesses**

From a purely technical perspective, the architectural innovation is limited. The segmentation rule based on acceleration zero crossings is simple, and the masked modeling objective follows standard SSL paradigms. The novelty lies primarily in the representation-level inductive bias rather than new modeling machinery.

Additionally, while transfer performance across multiple datasets is demonstrated, it remains unclear how broadly the submovement-based segmentation generalizes beyond wrist-worn IMU signals. Since submovement theory is rooted in point-to-point motor control, its applicability to other sensing locations or more complex motion regimes may warrant further investigation. The diversity and demographic breadth of the datasets may also influence how universally the proposed tokenization strategy applies.

---

> ### Author Rebuttal · Authors · 2026-03-30
>
> We thank the reviewer for recognizing that our tokenization is novel and well articulated, and that our evaluation is thorough.
>
>
> **Generalization beyond point-to-point motions:** While submovement theory was originally investigated using discrete, point-to-point wrist motions—which allow for the isolation of single submovements—the theory posits that these fundamental units can be combined to generate a vast repertoire of complex movements (see Hogan & Sternad, 2012). Consequently, our tokenization is not limited to discrete point-to-point motions but generalizes to the varied and naturalistic movement patterns captured in the six benchmarks. We demonstrate the empirical validity of this generalization through our superior performance across all datasets, including those featuring highly dynamic and non-linear activities. These clarifying points will be integrated into the revised manuscript.
>
>
> **Generalization to other sensing locations:** We appreciate the reviewer for highlighting this point and allowing us to clarify the broader applicability of our work. While this study validates the proposed tokenization specifically on wrist-worn accelerometer data, the approach is inherently extensible to other inertial modalities and body locations. The submovement theory by Hogan and Sternad applies not only to linear kinematics of the wrist but also more broadly to the linear/angular kinematics of various body joints (Hogan & Sternad, 2012); thus, it theoretically extends to gyroscope and magnetometer data, where angular and rotational velocity can be described as a superposition of bell-shaped primitives. Furthermore, while we focused on wrist-worn sensing—the most common and preferred configuration for long-term monitoring—prior evidence indicates that submovement theory applies to other end-effectors, including the chest, head, and ankles (Michmizos et al., 2014; PMID: 25505881, Chen et al., 2012; PMID: 23139749, de Lemos Fonseca et al., 2020; PMID: 31846518, Silva et al., 2026; j.bspc.2026.110099). We will include this discussion in the revised manuscript to highlight the broader impact and potential future directions of our work.
>
> ---
>
> **Performance gains across activity types:** We present confusion matrices for per-class analysis comparing against TF-C, the strongest baseline (Appendix H, Figures 8–12). Our major findings include:
>
> (1) *Transitional activities.* Our model shows substantial improvement in detecting transitional activities such as stand-to-sit/sit-to-stand (WHARF) and stand-to-lying/lying-to-stand. This underscores the model's ability to more precisely differentiate shifts in activity state by capturing the underlying sequential arrangement of constituent motions.
>
> (2) *Fine-Grained, similar activities.* The proposed tokenization significantly enhances the model's ability to distinguish between fine-grained, similar activity groups with overlapping signal characteristics (stairs vs. flat walking; tennis serve vs. right-hand throw; sitting vs. standing vs. lying; various eating/drinking gestures in MHealth/WISDM/WHARF). By shifting the learning objective from raw time-series morphology to the organizational structure of movement segments, we believe our model overcomes the traditional challenges in differentiating activities that share similar inertial signatures.
>
> **Impact of segmentation quality:** Since ground-truth labels for movement segments are unavailable, we cannot directly measure segmentation quality across different benchmarks. Instead, we evaluated how signal noise—which inherently degrades segmentation—affects downstream HAR performance. (For a detailed analysis, please see our response to Reviewer 2QTy).
>
> In short, our model remains highly robust to segmentation noise. This resilience is supported by our two-stage hysteresis filter, which mitigates spurious tokens and enables the Transformer to focus on the compositional structure of the most prominent movement segments. We will clarify these in the revised manuscript.
>
> ---
>
> **Limitations:** We agree that because the NHANES corpus is representative of the US population, the movement "grammars" learned by Bio-PM for functional activities may inherently reflect a bias toward Western or US-specific motor patterns.
>
> Furthermore, since motor control patterns are highly individual, it is important to acknowledge the privacy risk that our pipeline could pose. We will clarify these in the revised manuscript.
>
> ---
>
> We sincerely thank the reviewer for their thoughtful review; their insightful questions regarding broader applicability and per-class performance have directly strengthened our manuscript.

---

> > ### Author Rebuttal · Reviewer_VL8G · 2026-04-01
> >
> > The authors have thoroughly addressed the questions raised in the review. They provide clear clarification on the generalization of submovement-based tokenization beyond point-to-point motions, including its compositional nature and applicability across diverse activity types and sensing configurations. The discussion of performance across activity types and robustness to segmentation noise directly strengthens the empirical interpretation of the results. Additionally, the limitations regarding demographic bias and potential privacy considerations have been appropriately acknowledged. These responses fully resolve the concerns raised.

---

### Decision · Program_Chairs · 2026-04-30

**Decision:**

Accept (regular)

**Comment:**

This paper proposes a self-supervision framework to pretraining models that use wrist-worn IMU data for human activity recognition. They introduce a novel tokenization approach based on biologically inspired “submovements”, which is used for pretraining via masked reconstruction. The approach is evaluated on several datasets and compared against different SSL techniques, achieving superior performance. The reviewers agree that the proposed method is novel, well grounded and potentially significant, that the experimental evaluation is rigorous and thorough, and that the results support the claims.